# Revisiting Distribution Correction Estimation for Offline Imitation Learning with Suboptimal Dataset

Quang Anh Pham [1]   Tien Mai [1]   Akshat Kumar [1]

## Abstract

Imitation Learning (IL) learns high-quality policies from expert demonstrations but degrades in low-expert-data regimes. To address this, recent work studies "*offline IL with supplementary data*", augmenting expert data with trajectories from suboptimal policies. A prominent framework is Distribution Correction Estimation (DICE), which estimates density ratios via the dual of a divergence minimization problem between learned and expert visitation distributions. However, existing DICE-based methods either rely on a strict coverage assumption or introduce additional dataset regularization, limiting performance. We propose **ReDICE**, a new DICE-based method that addresses these issues through an objective-level reformulation. Our approach constructs a mixture-distribution objective that preserves the original expert-imitation objective while removing the coverage assumption, and its dual reduces to a stable Gumbel regression objective for efficient optimization. We further introduce a novel policy extraction mechanism that improves performance. Experiments on standard and real-world offline IL benchmarks show that ReDICE consistently outperforms prior methods and achieves state-of-the-art results.

## 1. Introduction

Offline IL Learning (IL) aims to learn decision-making policies by mimicking expert behavior from demonstrations, without requiring explicit reward specification or online interaction with the environment (Zare et al., 2024). This paradigm has proven effective across a wide range of sequential decision-making tasks, particularly in safety-critical

domains such as robotics and autonomous driving, where exploration is expensive or infeasible (Cheng et al., 2024; Black et al., 2025). The most widely used IL method is Behavior Cloning (BC) (Pomerleau, 1988), which reduces policy learning to supervised learning on expert state–action pairs. Despite its simplicity, BC is fundamentally limited by compounding errors (Ross et al., 2011), where minor execution deviations can lead the agent into unseen states from which it may not recover. To overcome this limitation, recent works (Xu et al., 2022; Li et al., 2023b) have studied *offline IL with suboptimal demonstrations* setting in which a small set of expert demonstrations is supplemented with a much larger dataset collected by suboptimal behavior policies. Such supplementary data are often inexpensive to obtain and can significantly expand state–action coverage. However, directly leveraging these data remains challenging, since naively imitating suboptimal behaviors can degrade policy performance.

A prominent class of methods addressing this setting is Distribution Correction Estimation (DICE) (Nachum et al., 2019), which formulates imitation learning as the minimization of a divergence between the occupancy measures of the learned policy and expert. Despite their strong theoretical foundations, existing DICE formulations suffer from important practical and conceptual limitations. The first DICE method, DemoDICE (Kim et al., 2021) modifies the original divergence objective by introducing an additional KL regularization term that penalizes deviation from the suboptimal data distribution. Although this regularization helps integrate suboptimal data into the learning process, it comes at the cost of no longer directly minimizing the divergence between the learned policy and the expert. Moreover, the strength of the regularization must be carefully tuned, and overly conservative regularization can prevent effective policy improvement when supplementary data are low-quality (Yu et al., 2023). Another approach, SMODICE (Ma et al., 2022) directly minimizes the KL divergence between the learned policy occupancy and the expert occupancy by decomposing the divergence using the suboptimal data distribution as a reference measure. However, this decomposition is only valid under a strict coverage assumption—the suboptimal dataset assigns non-zero probability to all state–action pairs visited by the expert. In practice, violations of this

---

[1]Singapore Management University, Singapore. Correspondence to: Quang Anh Pham <qa.pham.2025@phdcs.smu.edu.sg>.

*Proceedings of the 43rd International Conference on Machine Learning*, Seoul, South Korea. PMLR 306, 2026. Copyright 2026 by the author(s).

assumption can often lead to unstable optimization and poor performance (Sikchi et al., 2024).

These observations reveal a fundamental limitation of existing DICE-based methods: enforcing expert coverage through suboptimal data can lead to biased or unstable objectives, while adding regularization compromises the original imitation learning objective. This raises a natural question: **Is it possible to design a DICE objective that remains well-defined without coverage assumptions while still directly optimizing expert imitation?** We answer this question in this paper by proposing ReDICE (**Re**visiting **D**istribution **C**orrection **E**stimation), a new offline IL learning method that resolves these issues. Specifically, our contributions are as follows:

- We propose a mixture-based KL divergence formulation for offline IL, which directly minimizes the divergence between the learned policy occupancy and the expert occupancy. This formulation removes the strict expert coverage assumption and avoids introducing auxiliary regularization terms, while preserving a principled divergence minimization objective.

- We derive a dual representation of the proposed objective via Lagrangian relaxation of Bellman flow constraints and show that the resulting dual objective is equivalent to a Gumbel regression loss (Garg et al., 2023). This loss is strictly convex with a unique global optimum that helps stabilize the learning process and improve convergence. Building on this dual formulation, we develop a practical policy extraction mechanism that mitigates weight collapse caused by value overestimation, while remaining consistent with the theoretical solution.

- We evaluate ReDICE through extensive experiments on standard offline IL benchmarks with different levels of expert coverage, and a real-world maritime navigation setting. Experimental results show that ReDICE outperforms state-of-the-art baselines. The implementation of ReDICE is publicly available at `https://github.com/quanganh1999/ReDICE`.

## 2. Related Work

**Offline IL with suboptimal data** Learning solely from expert demonstrations often requires a large amount of expert data, which can be costly or difficult to obtain in practice. To address this, prior work has studied offline imitation learning with suboptimal data, where limited expert demonstrations are augmented with inexpensive but imperfect trajectories. DemoDICE (Kim et al., 2021) extends the BC objective by introducing a KL-divergence regularization term that constrains the learned policy to remain close to the behavior policy. However, its performance can degrade when the supplementary data are highly suboptimal. Subsequent approaches mitigate this issue by proposing different variants of weighted BC. DWBC (Xu et al., 2022) learns a discriminator to assign importance weights to demonstrations, while ISWBC (Li et al., 2023b) reweights the BC loss using a density ratio between expert and offline data to prioritize expert-like samples. ILID (Yue et al., 2024) further leverages environment dynamics to identify beneficial behaviors that lead the agent back to known expert states, thereby empowering the policy to recover from deviations using imperfect but informative demonstrations. SMODICE (Ma et al., 2022) considers a related setting in which the expert dataset contains only states without actions. Unlike DemoDICE, SMODICE directly minimizes the KL divergence between the learned policy occupancy and the expert occupancy using offline data. While theoretically appealing, this approach relies on a restrictive coverage assumption that the suboptimal data cover expert visitations. ReCOIL (Sikchi et al., 2024) relaxes this assumption by optimizing alternative divergence minimization between the two mixture occupancy measures. While this modification improves performance in practice, ReCOIL does not guarantee recovery of the optimal density ratio corresponding to the original expert–policy divergence objective. We provide a detailed comparison between ReCOIL and our formulation in Appendix B, showing that ReDICE preserves equivalence to the original KL-based imitation objective while avoiding the max–min optimization and additional surrogate used in ReCOIL.

**DICE-based Methods** Stationary distribution correction estimation was first introduced by (Nachum et al., 2019) as the DICE framework, which enables estimation of the stationary distribution of a target policy using off-policy data. The key idea is to formulate learning objectives directly in terms of distribution ratios through duality and change-of-variables via importance sampling (Nachum & Dai, 2020). This framework has since been applied successfully across a variety of reinforcement learning (RL) problems: off-policy evaluation (Zhang et al., 2020; Yang et al., 2020), reward learning (Li et al., 2023a), offline RL (Lee et al., 2021; Mao et al., 2024), constrained RL (Lee et al., 2022), unsupervised RL (Lee et al., 2025), online IL (Kostrikov et al., 2020), and offline IL (Kim et al., 2021; Ma et al., 2022). ReDICE differs from prior DICE-based offline IL methods by using supplementary data as a mixture reference rather than as the direct imitation target or only as a behavior regularizer. This design relaxes the coverage limitations of expert-only matching while remaining equivalent to the original KL-based imitation objective.

## 3. Preliminaries

We consider a Markov Decision Process (MDP) denoted by the tuple $\mathcal{M} = (\mathcal{S}, \mathcal{A}, \mathcal{P}, r, \gamma, d_0)$, where $\mathcal{S}$ is the state space, $\mathcal{A}$ is the action space, $\mathcal{P}(s'|s, a)$ is the transition

dynamics, $r : \mathcal{S} \times \mathcal{A} \to \mathbb{R}$ is the reward function, $\gamma \in (0, 1)$ is the discount factor, and $d_0(s)$ is the initial state distribution. A stochastic policy $\pi : \mathcal{S} \to \Delta(\mathcal{A})$ maps each state to a distribution over actions. Executing the policy $\pi$ in MDP $\mathcal{M}$ induces a discounted state-action visitation distribution, known as the occupancy measure:

$$d^\pi(s, a) = (1 - \gamma) \sum_{t=0}^{\infty} \gamma^t \Pr(s_t = s, a_t = a \mid s_0 \sim d_0,$$
$$a_t \sim \pi(\cdot \mid s_t), \ s_{t+1} \sim \mathcal{P}(\cdot \mid s_t, a_t)). \quad (1)$$

We can represent the standard RL objective which finds a policy maximizing the expected return under its occupancy measure i.e. $\max_\pi \{\mathbb{E}_{s,a \sim d^\pi}[r(s, a)]\}$.

### 3.1. Offline RL with Distribution Correction Estimation

Assuming that we have access to a static dataset $\mathcal{D} = (s, a, r, s')$ generated by any behavior policy. The goal of DICE approaches (Nachum & Dai, 2020) is leveraging $\mathcal{D}$ to optimize $\max_\pi \{\mathbb{E}_{s,a \sim d^\pi}[r(s, a)]\}$, thus removing the need of on-policy samples from $d^\pi$. Specifically, they add a behavior regularization term into the original objective as:

$$\max_\pi \mathbb{E}_{s,a \sim d^\pi}[r(s, a)] - \alpha D_f(d^\pi(s, a) || d^D(s, a)) \quad (2)$$

where $D_f$ is an $f$-divergence between two distributions, and $d^D(s, a)$ denotes the visitation distribution of policy generating dataset $\mathcal{D}$. We then omit $d^\pi$ from (2) by rewriting it as an optimization problem over the space of valid state-action visitation distributions:

$$\max_{d \geq 0} \mathbb{E}_{s,a \sim d}[r(s, a)] - \alpha D_f(d(s, a) || d^D(s, a)) \quad (3)$$
$$\text{s.t.} \quad \sum_a d(s, a) = (1 - \gamma) d_0(s) + \gamma \mathcal{T}_\star d(s), \quad \forall s \quad (4)$$

where $\mathcal{T}_\star d(s) = \sum_{\overline{s}, \overline{a}} d(\overline{s}, \overline{a}) \mathcal{P}(s \mid \overline{s}, \overline{a})$, and (4) are *Bellman flow constraints* (Manne, 1960) ensuring that $d(s, a)$ is a valid occupancy measure.

Solving the primal problem in Eq. (3) directly is intractable due to the flow constraints. We convert it to an unconstrained formulation by applying Lagrangian duality (Detailed Derivation in Appendix A.1):

$$\min_V (1 - \gamma) \mathbb{E}_{s \sim d_0}[V(s)] + \alpha \mathbb{E}_{s,a \sim d^D}[f^*(\frac{\mathcal{T}V(s, a) - V(s)}{\alpha})] \quad (5)$$

where $f^*(x)$ is a variant of convex conjugate defined as $f^*(x) = \max\{0, (f')^{-1}(x)\} \cdot x - f(\max\{0, (f')^{-1}(x)\})$, and $\mathcal{T}V(s, a) = r(s, a) + \gamma \mathbb{E}_{s' \sim \mathcal{P}(.|s,a)}[V(s')]$ is a Bellman operator on $V$. The optimal solution $V^*$ of (5) can be used to compute the optimal density ratio (distribution correction):

$$\frac{d^*(s, a)}{d^D(s, a)} = \max\left(0, (f')^{-1}(\frac{\mathcal{T}V(s, a) - V(s)}{\alpha})\right) \quad (6)$$

### 3.2. DICE for Offline IL with Supplementary Data

In this setting, we are provided with two static datasets: an expert dataset $\mathcal{D}_E = \{(s, a, s')\}$ generated by an expert policy $\pi_E$, and a suboptimal dataset $\mathcal{D}_S = \{(s, a, s')\}$ generated by low-quality policies. We denote the stationary visitation distributions of these datasets as $d^E(s, a)$ and $d^S(s, a)$, respectively. The goal is to learn a policy $\pi$ that mimics the expert behavior using only $\mathcal{D}_E$ and $\mathcal{D}_S$, without environment interaction. We use the shorthand $D_{\mathrm{KL}}(p || q)$ for $D_{\mathrm{KL}}(p(s, a) || q(s, a))$ with some state-action visitation distributions $p$ and $q$ to simplify the notation. We consider two main DICE formulations introduced in previous works for the Offline IL with Supplementary Data: SMODICE (Ma et al., 2022) and DemoDICE (Kim et al., 2021).

Even though SMODICE focuses on minimizing the KL divergence between two state visitation distributions of policy and expert, we can easily extend its formulation to state-action visitation distribution as:

$$D_{\mathrm{KL}}(d^\pi(s, a) || d^E(s, a)) \quad (7)$$
$$= \mathbb{E}_{d^\pi}\left[\log\left(\frac{d^\pi(s, a)}{d^E(s, a)} \cdot \frac{d^S(s, a)}{d^S(s, a)}\right)\right] \quad (8)$$
$$= \mathbb{E}_{d^\pi}\left[\log\frac{d^S(s, a)}{d^E(s, a)}\right] + \mathbb{E}_{d^\pi}\left[\log\frac{d^\pi(s, a)}{d^S(s, a)}\right] \quad (9)$$
$$= \mathbb{E}_{d^\pi}\left[\log\frac{d^S(s, a)}{d^E(s, a)}\right] + D_{KL}(d || d^S) \quad (10)$$

SMODICE relies on the coverage assumption ($d^S(s, a) > 0$ whenever $d^E(s, a) > 0$) to ensure that the log decomposition (Step (8) to (9)) is valid. Based on (10), we can convert $\min_\pi D_{\mathrm{KL}}(d^\pi(s, a) || d^E(s, a))$ into the following maximization problem:

$$\max_\pi \mathbb{E}_{d^\pi}\left[-\log\frac{d^S(s, a)}{d^E(s, a)}\right] - D_{KL}(d || d^S) \quad (11)$$

(11) is exactly a DICE objective (2) if we set $r(s, a) = -\log\frac{d^S(s,a)}{d^E(s,a)}$, $\alpha = 1$, and $d^D = d^S$. To estimate $r(s, a)$, we train a discriminator $c : \mathcal{S} \times \mathcal{A} \to (0, 1)$ using GAN objective (Goodfellow et al., 2014):

$$\max_c \mathbb{E}_{d^E}[\log c(s, a)] + \mathbb{E}_{d^S}[\log(1 - c(s, a))] \quad (12)$$

The optimal discriminator $c^*(s, a) = \frac{d^E(s,a)}{d^E(s,a) + d^S(s,a)}$. Therefore, we can compute $r(s, a) = -\log(\frac{1}{c^*(s,a)} - 1)$

DemoDICE modifies the original divergence objective by introducing an additional KL regularization term that penalizes deviation from the suboptimal data distribution:

$$\min_\pi D_{\mathrm{KL}}(d^\pi || d^E) + \lambda D_{\mathrm{KL}}(d^\pi || d^S) \quad (13)$$

Similar to SMODICE, they require the coverage assumption to decompose $D_{KL}(d^\pi(s,a) \| d^E(s,a))$ to obtain a DICE objective:

$$\max_\pi \mathbb{E}_{d^\pi}\left[ -\log \frac{d^S(s,a)}{d^E(s,a)} \right] - (1+\lambda)D_{KL}(d\|d^S) \quad (14)$$

We can see that the primary limitation of both SMODICE and DemoDICE is reliance on the coverage assumption. DemoDICE does not directly minimize expert–policy divergence and depends on a KL regularization term controlled by the hyperparameter $\lambda$.

## 4. Methodology

In this section, we present ReDICE, a novel offline IL method to deal with the coverage-assumption issue of previous DICE works while still directly optimizing the KL divergence between policy and expert.

### 4.1. Relaxing Coverage Assumption via Mixture Distribution

Different from prior works, we decompose the KL divergence $D_{KL}(d^\pi \| d^E)$ using a mixture distribution $d^{mix} = \beta d^E + (1-\beta)d^S$ with $\beta \in (0,1)$ as follows:

$$D_{KL}(d^\pi(s,a) \| d^E(s,a)) \quad (15)$$

$$= \mathbb{E}_{d^\pi}\left[ \log\left( \frac{d^\pi(s,a)}{d^E(s,a)} \cdot \frac{d^{mix}(s,a)}{d^{mix}(s,a)} \right) \right] \quad (16)$$

$$= \mathbb{E}_{d^\pi}\left[ \log \frac{d^{mix}(s,a)}{d^E(s,a)} \right] + \mathbb{E}_{d^\pi}\left[ \log \frac{d^\pi(s,a)}{d^{mix}(s,a)} \right] \quad (17)$$

This decomposition does not require the coverage assumption because $d^{mix}(s,a)$ is always bigger than 0 whenever $d^E(s,a) > 0$. Therefore, minimizing the original KL divergence is equivalent to optimizing the following DICE objective:

$$\max_{d \geq 0} \mathbb{E}_d[r(s,a)] - D_{KL}(d(s,a)\|d^{mix}(s,a)) \quad (18)$$

$$\text{s.t.} \quad \text{Constraints (4)}$$

where the reward function $r(s,a)$ is defined as $r(s,a) = -\log\left( \frac{d^{mix}(s,a)}{d^E(s,a)} \right)$. To estimate this reward function, we can estimate the density ratio $\frac{\beta d^E(s,a)}{d^{mix}(s,a)} = c^*(s,a)$, which corresponds to the optimal solution of the following discriminator training objective:

$$\max_c \beta \mathbb{E}_{d^E}[\log c(s,a)] + (1-\beta)\mathbb{E}_{d^S}\left[\log\big(1 - c(s,a)\big)\right]. \quad (19)$$

Then the reward function can then be recovered as $r(s,a) = -\log \frac{\beta}{c^*(s,a)}$.

### 4.2. Dual Optimization via Gumbel Regression

As described in Section 3.1, optimizing the objective (18) with flow constraints can be done via solving its dual formulation:

$$\min_V (1-\gamma)\mathbb{E}_{s \sim d_0}[V(s)] + \mathbb{E}_{d^{mix}}[f^*(\mathcal{T}V(s,a) - V(s))] \quad (20)$$

where $f^*(x) = \max\{0, (f')^{-1}(x)\} \cdot x - f\big(\max\{0, (f')^{-1}(x)\}\big)$ and $\mathcal{T}V(s,a) = r(s,a) + \gamma\mathbb{E}_{s' \sim \mathcal{P}(.|s,a)}[V(s')]$.

For KL divergence, $f(x)$ can be $x\log x$, or $x\log x - x + 1$. (Schulman, 2020) empirically shows that the latter has lower variance than the former. Therefore, we use $f(x) = x\log x - x + 1$ which leads to $(f')^{-1}(x) = e^x$ and

$$f^*(x) = (\max\{0, e^x\})\,x - f\left(\max\{0, e^x\}\right) = e^x - 1$$

in our formulation. Furthermore, we propose to optimize this objective using a *Gumbel regression loss*: (Garg et al., 2023). We leverage the following result:

**Proposition 4.1.** *Optimizing the dual objective (20) under KL divergence is equivalent to minimizing the following Gumbel regression loss:*

$$\min_V \mathbb{E}_{(s,a) \sim d^{mix}}\left[ e^{\mathcal{T}V(s,a) - V(s)} \right]$$
$$- \mathbb{E}_{(s,a) \sim d^{mix}}\left[ \mathcal{T}V(s,a) - V(s) \right] - 1 \quad (21)$$

Optimizing (21) offers several advantages. First, it avoids the need to estimate $(1-\gamma)\mathbb{E}_{d_0}[V(s)]$, whereas both SMODICE and DemoDICE require sampling initial states from the dataset for this purpose. This requirement is not always feasible when initial-state data are limited or unavailable. A common workaround is to replace $d_0$ with $d^S$ (Sikchi et al., 2024), which may work empirically but leads to biased estimation. Finally, the objective in (21) is strictly convex in $V$, guaranteeing a unique minimum in the V-space and thereby improving optimization stability and convergence during learning.

### 4.3. Policy Extraction

After learning the optimal value function $V^*(s)$, we extract the policy. Theoretically, the optimal policy can be extracted via the following weighted behavior cloning:

$$\max_\pi \mathbb{E}_{d^{mix}}\left[w(s,a)\log\pi(a|s)\right] \quad (22)$$

where $w(s,a) = \frac{d^{\pi^*}(s,a)}{d^{mix}(s,a)}$ is the optimal distribution correction defined as follows:

$$w(s,a) = \max\big(0, (f')^{-1}\big(\mathcal{T}V(s,a) - V(s)\big)\big)$$
$$= \max\big(0, e^{\mathcal{T}V(s,a) - V(s)}\big) = e^{\mathcal{T}V(s,a) - V(s)}$$

However, we find that directly using this weight does not result in good empirical performance. One possible explanation is that $V(s)$ can be overestimated, which causes the exponential weight to become negligibly small and thus suppress potentially useful data points. To mitigate this issue, we propose an alternative objective:

**Proposition 4.2.** *An optimal solution to the policy extraction (22) is also an optimal solution to the following learning objective:*

$$\max_{\pi} \mathbb{E}_{s,a \sim d^{mix}} \left[ e^{\mathcal{T}V(s,a) - \tau V(s)} \log \pi(a|s) \right] \quad (23)$$

*where $\tau \in [0,1]$ is a scalar hyperparameter.*

Setting $\tau < 1$ effectively reduces the influence of the value function on the current state, acting as a regularizer.

### 4.4. Practical Implementation

Computing the Bellman operator $\mathcal{T}V(s,a) = r(s,a) + \gamma \mathbb{E}_{s' \sim \mathcal{P}(.|s,a)}[V(s')]$ requires the knowledge of transition dynamics $\mathcal{P}$ which is not available in offline setting. We follow previous works (Sikchi et al., 2024; Mao et al., 2024) using an empirical Bellman operator that only utilizes a single $s'$ sample from datasets. Specifically, we optimize the following objective to learn the value function $V(s)$:

$$
\begin{aligned}
\min_{V} \mathcal{L}(V) = \; &\mathbb{E}_{(s,a,s') \sim d^{mix}} \left[ e^{r(s,a) + V(s') - V(s)} \right] \\
&- \mathbb{E}_{(s,a,s') \sim d^{mix}} \left[ r(s,a) + V(s') - V(s) \right] - 1 \quad (24)
\end{aligned}
$$

Note that, (21) and (24) are equivalent under deterministic transitions.

Optimizing (24) directly faces instability when trained via gradient descent. This instability can be understood as arising from conflicting gradient signals associated with $V(s)$ and $V(s')$, particularly when nearby states share similar learned feature representations due to feature co-adaptation (Kumar et al., 2022). To overcome this limitation, previous works (Xu et al., 2023; Sikchi et al., 2024) often use a semi-gradient update rule which avoids differentiating through the $V(s')$. Specifically, they incorporate another value function $Q(s,a)$ denoting a stop-gradient operator of $r(s,a) + V(s')$ and optimize $Q$ via a MSE (mean squared error) loss. However, this increases the training cost due to adding another neural network. We instead adopt the orthogonal gradient update introduced in ODICE (Mao et al., 2024) for offline RL, which resolves gradient interference by decomposing the update into a projection of the gradient of and its orthogonal complement in a principled manner. We refer readers to ODICE for a detailed discussion. The full practical algorithm is summarized in Algorithm 1.

---

**Algorithm 1** ReDICE

---

**Input:** Expert dataset $\mathcal{D}_E$, Suboptimal dataset $\mathcal{D}_S$, Hyperparameters $\gamma, \tau, \beta$.
Initialize value, policy and discriminator networks $V_\omega, \pi_\theta, c_\phi$.
Train discriminator $c_\phi$ via (19)
Compute rewards $r(s,a) = -\log \frac{\beta}{c_\phi(s,a)} \; \forall (s,a) \in \mathcal{D}_E \cup \mathcal{D}_S$
**for** step $t = 1$ to $T$ **do**
    Sample mini-batch $B = \{(s,a,s')\} \sim \mathcal{D}_E \cup \mathcal{D}_S$.
    **Update Value Function:** Train $V_\omega$ via Gumbel loss (Eq. 24) using orthogonal gradient update.
    **Policy Extraction:** Update policy $\pi_\theta$

$$\max_{\theta} \mathbb{E}_{d^{mix}} \left[ e^{r(s,a) + \gamma V(s') - \tau V(s)} \log \pi_\theta(a|s) \right]$$

**end for**

---

## 5. Experiments

We empirically address following questions: **(Q1)** How well does ReDICE perform compared to prior state-of-the-art methods on standard offline IL benchmarks? **(Q2)** What is the performance of ReDICE on a limited expert coverage setting? **(Q3)** Can we apply ReDICE to a real-world scenario? **(Q4)** What are the effects of the proposed components, and key hyperparameters on ReDICE performance? The environment and implementation details are in the Appendix C.

**Baselines and Experimental Setup** We compare our method against several representative IL baselines. **BCE** trains a BC policy solely on expert demonstrations, while **BCU** extends BCE by training on both expert and suboptimal datasets. For DICE methods, we include **DemoDICE** (Kim et al., 2021) and **SMODICE-action**, which adapts SMODICE (Ma et al., 2022) to leverage action information. We also compare against **ReCOIL** (Sikchi et al., 2024), a state-of-the-art offline IL method that outperforms SMODICE; and **ILID** (Yue et al., 2024), a discriminator-weighted BC approach that has demonstrated strong performance in the challenging setting where suboptimal data have low expert coverage. Each method is trained for 1M gradient steps across 5 random seeds, with performance evaluated periodically using the *normalized score* = $100 * \frac{\text{method score - random score}}{\text{expert score - random score}}$ (Fu et al., 2020). For comparison, we report the average normalized score computed from the final 10 evaluations.

### 5.1. Offline IL

To address (Q1), we adopt the same offline IL benchmark used in ReCOIL (Sikchi et al., 2024), with datasets constructed from the D4RL framework (Fu et al., 2020). We evaluate all methods on eight MuJoCo (Todorov et al., 2012) environments, including four locomotion (*Hopper, HalfCheetah, Walker2d, Ant*) and four manipulation (*Pen,*

*Table 1.* Average normalized return over last 10 evaluations of ReDICE against baselines on the D4RL suboptimal datasets with 1 expert trajectory. The mean and std are obtained over 5 random seeds. Offline IL methods with avg. perf within the std-dev of the top performing approach is in **bold**.

| Suboptimal Dataset | Env | BCE | BCU | DemoDICE | SMODICE-action | ILID | ReCOIL | ReDICE | Expert |
|---|---|---|---|---|---|---|---|---|---|
| random+ | hopper | $17.87_{\pm1.66}$ | $4.31_{\pm0.17}$ | $99.85_{\pm1.98}$ | $104.09_{\pm1.82}$ | $99.57_{\pm10.58}$ | $98.33_{\pm12.97}$ | $\mathbf{109.80_{\pm0.31}}$ | 111.3 |
| | halfcheetah | $2.46_{\pm0.45}$ | $4.06_{\pm0.93}$ | $81.97_{\pm3.84}$ | $80.92_{\pm1.94}$ | $3.50_{\pm1.07}$ | $78.78_{\pm6.71}$ | $\mathbf{91.82_{\pm0.44}}$ | 88.83 |
| expert | walker2d | $11.42_{\pm5.22}$ | $1.40_{\pm0.13}$ | $\mathbf{107.89_{\pm0.24}}$ | $107.46_{\pm0.22}$ | $107.81_{\pm0.23}$ | $105.72_{\pm3.55}$ | $\mathbf{107.98_{\pm0.12}}$ | 106.92 |
| | ant | $24.74_{\pm1.47}$ | $37.82_{\pm5.05}$ | $\mathbf{126.80_{\pm0.71}}$ | $126.48_{\pm1.78}$ | $90.72_{\pm4.52}$ | $116.43_{\pm3.69}$ | $\mathbf{126.32_{\pm0.81}}$ | 130.75 |
| random+ | hopper | $17.87_{\pm1.66}$ | $4.11_{\pm0.13}$ | $78.15_{\pm8.40}$ | $58.72_{\pm7.09}$ | $\mathbf{99.79_{\pm14.15}}$ | $86.65_{\pm20.85}$ | $\mathbf{93.63_{\pm1.69}}$ | 111.33 |
| | halfcheetah | $2.46_{\pm0.45}$ | $2.24_{\pm0.02}$ | $16.63_{\pm6.75}$ | $4.16_{\pm2.83}$ | $2.22_{\pm0.02}$ | $5.25_{\pm0.49}$ | $\mathbf{79.31_{\pm4.59}}$ | 88.83 |
| few-expert | walker2d | $11.42_{\pm5.22}$ | $1.07_{\pm0.24}$ | $99.54_{\pm8.26}$ | $3.48_{\pm2.57}$ | $106.87_{\pm1.42}$ | $12.59_{\pm11.04}$ | $\mathbf{108.26_{\pm0.03}}$ | 106.92 |
| | ant | $24.74_{\pm1.47}$ | $37.73_{\pm6.58}$ | $89.11_{\pm6.83}$ | $13.26_{\pm5.01}$ | $82.34_{\pm7.25}$ | $56.68_{\pm15.22}$ | $\mathbf{108.15_{\pm3.59}}$ | 130.75 |
| medium+ | hopper | $17.87_{\pm1.66}$ | $54.69_{\pm0.45}$ | $81.23_{\pm12.90}$ | $57.83_{\pm3.52}$ | $34.10_{\pm13.25}$ | $85.25_{\pm20.86}$ | $\mathbf{111.01_{\pm0.38}}$ | 111.33 |
| | halfcheetah | $2.46_{\pm0.45}$ | $42.61_{\pm0.20}$ | $75.59_{\pm3.56}$ | $66.52_{\pm2.32}$ | $43.26_{\pm1.65}$ | $78.88_{\pm3.56}$ | $\mathbf{91.80_{\pm0.42}}$ | 88.83 |
| expert | walker2d | $11.42_{\pm5.22}$ | $73.43_{\pm1.66}$ | $106.65_{\pm0.64}$ | $39.71_{\pm10.78}$ | $75.56_{\pm51.32}$ | $108.27_{\pm1.46}$ | $\mathbf{108.40_{\pm0.50}}$ | 106.92 |
| | ant | $24.74_{\pm1.47}$ | $97.37_{\pm4.08}$ | $114.56_{\pm4.50}$ | $109.73_{\pm1.88}$ | $73.30_{\pm16.61}$ | $119.21_{\pm3.71}$ | $\mathbf{123.24_{\pm1.48}}$ | 130.75 |
| medium+ | hopper | $17.87_{\pm1.66}$ | $52.51_{\pm0.52}$ | $61.65_{\pm1.97}$ | $53.33_{\pm0.80}$ | $20.98_{\pm6.43}$ | $38.88_{\pm25.99}$ | $\mathbf{109.50_{\pm1.23}}$ | 111.33 |
| | halfcheetah | $2.46_{\pm0.45}$ | $42.52_{\pm0.28}$ | $47.41_{\pm2.99}$ | $41.47_{\pm1.99}$ | $42.67_{\pm1.21}$ | $41.90_{\pm3.37}$ | $\mathbf{81.57_{\pm2.94}}$ | 88.83 |
| few-expert | walker2d | $11.42_{\pm5.22}$ | $72.26_{\pm1.48}$ | $87.02_{\pm6.02}$ | $13.78_{\pm14.21}$ | $82.67_{\pm9.96}$ | $71.44_{\pm15.45}$ | $\mathbf{108.70_{\pm0.44}}$ | 106.92 |
| | ant | $24.74_{\pm1.47}$ | $88.81_{\pm0.54}$ | $97.71_{\pm0.85}$ | $89.22_{\pm2.44}$ | $71.08_{\pm10.07}$ | $104.03_{\pm4.74}$ | $\mathbf{111.73_{\pm2.37}}$ | 130.75 |
| cloned+expert | pen | $8.84_{\pm1.64}$ | $25.75_{\pm3.23}$ | $36.91_{\pm5.75}$ | $33.97_{\pm19.99}$ | $7.07_{\pm5.70}$ | $45.59_{\pm9.17}$ | $\mathbf{88.84_{\pm8.84}}$ | 106.42 |
| | door | $0.02_{\pm0.07}$ | $0.13_{\pm0.20}$ | $0.07_{\pm0.07}$ | $3.57_{\pm3.38}$ | $-0.03_{\pm0.04}$ | $16.18_{\pm17.29}$ | $\mathbf{92.23_{\pm2.45}}$ | 103.94 |
| | hammer | $0.32_{\pm0.14}$ | $4.29_{\pm3.72}$ | $0.91_{\pm0.74}$ | $3.98_{\pm5.45}$ | $0.19_{\pm0.11}$ | $2.60_{\pm1.35}$ | $\mathbf{104.86_{\pm10.07}}$ | 125.71 |
| human+expert | pen | $8.84_{\pm1.64}$ | $27.07_{\pm3.91}$ | $39.70_{\pm8.36}$ | $46.08_{\pm3.85}$ | $11.65_{\pm7.85}$ | $\mathbf{85.04_{\pm5.56}}$ | $82.62_{\pm3.58}$ | 106.42 |
| | door | $0.02_{\pm0.07}$ | $8.77_{\pm6.73}$ | $18.37_{\pm7.47}$ | $26.16_{\pm11.76}$ | $-0.05_{\pm0.02}$ | $84.02_{\pm11.42}$ | $\mathbf{94.31_{\pm1.83}}$ | 103.94 |
| | hammer | $0.32_{\pm0.14}$ | $30.37_{\pm14.03}$ | $27.38_{\pm5.80}$ | $49.13_{\pm13.31}$ | $1.56_{\pm2.53}$ | $86.45_{\pm14.59}$ | $\mathbf{107.97_{\pm4.85}}$ | 125.71 |
| partial+expert | kitchen | $4.58_{\pm6.23}$ | $36.25_{\pm2.50}$ | $46.67_{\pm2.55}$ | $38.33_{\pm7.64}$ | $45.50_{\pm6.38}$ | $48.80_{\pm4.00}$ | $\mathbf{54.00_{\pm1.32}}$ | 75.00 |
| mixed+expert | kitchen | $4.58_{\pm6.23}$ | $44.50_{\pm1.39}$ | $45.67_{\pm2.75}$ | $46.67_{\pm5.20}$ | $2.58_{\pm4.26}$ | $49.35_{\pm1.66}$ | $48.25_{\pm0.35}$ | 75.00 |
| Average | | 10.56 | 33.09 | 66.14 | 50.75 | 46.04 | 67.76 | **97.68** | 107.23 |

*Door, Hammer, Kitchen*). For each task, the expert dataset consists of a single trajectory. In the locomotion domains, suboptimal datasets are formed by combining D4RL random or medium data with either 200 expert trajectories (*expert*) or 30 expert trajectories (*few-expert*). For manipulation tasks, non-expert datasets from D4RL (mixed and partial for Kitchen; human and cloned for the remaining tasks) are combined with up to 30 expert trajectories. This construction yields 24 diverse tasks for benchmarking. Manipulation and 'few-expert' tasks pose greater challenges due to their higher-dimensional state spaces and low expert coverage. Additional details on environments and dataset construction are provided in the Appendix C.1.

Table 1 summarizes the average normalized score of ReDICE and prior offline IL baselines. Overall, ReDICE achieves the best aggregate performance (97.68 average), outperforming the strongest baseline on average by about 44% (vs. ReCOIL: 67.76), and reaching 91.1% of the expert average (107.23). ReDICE is the top-performing method on all tasks. The gains are especially considerable in the few-expert setting, where several baselines collapse (e.g., on halfcheetah under random+few-expert), while ReDICE retains strong near-expert performance. On the more challenging manipulation tasks like pen, door and hammer, ReDICE substantially outperforms prior methods.

## 5.2. Offline IL with Limited Expert Coverage

To address (Q2), we consider an even more challenging setting with *very low expert coverage*, similar to the setup studied in ILID (Yue et al., 2024). The expert dataset still contains only a single trajectory, but unlike the benchmark in Section 5.1, the suboptimal dataset is *not* mixed with any expert rollouts and is instead collected purely from low-quality behaviors (e.g., D4RL random/medium or cloned/human). This setting stresses an algorithm's ability to correctly propagate sparse expert information.

Table 2 reports normalized scores for this setting. We additionally include an **OfflineRL** column as an approximate upper bound, constructed from the best score among multiple offline RL methods reported in ReBRAC (Tarasov et al., 2023) when trained on the same suboptimal datasets with access to ground-truth rewards. For Ant tasks that are not reported in ReBRAC, we run ReBRAC ourselves using a sweep of configurations adapted from the other locomotion tasks and report the best score. Overall, ReDICE is close to methods with access to true rewards, while consistently outperforming prior offline IL baselines. The gains are particularly large on tasks where other DICE-based methods struggle (e.g., HalfCheetah and manipulation tasks). These results clearly demonstrate that ReDICE does not rely on the coverage assumption and can therefore learn from diverse behavior in the suboptimal dataset.

*Table 2.* Average normalized score over last 10 evaluations of ReDICE against baselines on the *limited expert coverage* setting, with 1 expert trajectory and *no expert rollouts mixed into the suboptimal dataset*. The mean and std are obtained over 5 random seeds. Offline IL methods with avg. perf within the std-dev of the top performing approach is in **bold**.

| Suboptimal Dataset | Env | BCE | BCU | DemoDICE | SMODICE-action | ILID | ReCOIL | ReDICE || OfflineRL |
|---|---|---|---|---|---|---|---|---|---|
| random | hopper | $17.87_{\pm1.66}$ | $2.47_{\pm0.33}$ | $4.94_{\pm0.42}$ | $17.77_{\pm7.53}$ | $\mathbf{23.49_{\pm15.37}}$ | $\mathbf{19.97_{\pm12.62}}$ | $\mathbf{25.36_{\pm7.23}}$ | 19.6 |
| | halfcheetah | $2.46_{\pm0.45}$ | $2.24_{\pm0.00}$ | $2.21_{\pm0.02}$ | $2.15_{\pm0.06}$ | $2.25_{\pm0.01}$ | $2.28_{\pm0.23}$ | $\mathbf{16.79_{\pm1.13}}$ | 31.1 |
| | walker2d | $11.42_{\pm5.22}$ | $0.70_{\pm0.23}$ | $10.73_{\pm2.88}$ | $-0.25_{\pm0.19}$ | $\mathbf{17.04_{\pm14.10}}$ | $6.33_{\pm0.64}$ | $\mathbf{15.59_{\pm1.44}}$ | 18.7 |
| | ant | $24.74_{\pm1.47}$ | $30.74_{\pm0.53}$ | $27.48_{\pm1.15}$ | $27.05_{\pm3.27}$ | $30.90_{\pm3.18}$ | $31.50_{\pm0.03}$ | $\mathbf{36.65_{\pm3.35}}$ | 49.84 |
| medium | hopper | $17.87_{\pm1.66}$ | $52.45_{\pm1.42}$ | $70.73_{\pm3.68}$ | $53.99_{\pm0.79}$ | $23.06_{\pm8.00}$ | $36.10_{\pm8.89}$ | $\mathbf{82.13_{\pm2.54}}$ | 102 |
| | halfcheetah | $2.46_{\pm0.45}$ | $42.44_{\pm0.22}$ | $41.20_{\pm0.58}$ | $40.59_{\pm0.51}$ | $42.76_{\pm0.54}$ | $35.06_{\pm1.04}$ | $\mathbf{49.58_{\pm0.28}}$ | 66.4 |
| | walker2d | $11.42_{\pm5.22}$ | $64.86_{\pm4.50}$ | $\mathbf{70.73_{\pm3.68}}$ | $3.94_{\pm5.29}$ | $67.69_{\pm2.42}$ | $61.16_{\pm2.55}$ | $66.22_{\pm2.97}$ | 92.7 |
| | ant | $24.74_{\pm1.47}$ | $89.80_{\pm2.35}$ | $70.73_{\pm3.68}$ | $-32.07_{\pm25.45}$ | $84.38_{\pm4.14}$ | $92.17_{\pm2.99}$ | $\mathbf{99.05_{\pm1.04}}$ | 133.31 |
| cloned | pen | $8.84_{\pm1.64}$ | $22.91_{\pm3.99}$ | $29.76_{\pm0.98}$ | $14.16_{\pm21.06}$ | $2.76_{\pm8.70}$ | $30.67_{\pm2.38}$ | $\mathbf{31.65_{\pm1.58}}$ | 91.8 |
| | door | $0.02_{\pm0.07}$ | $0.01_{\pm0.00}$ | $0.17_{\pm0.04}$ | $0.94_{\pm1.54}$ | $-0.11_{\pm0.10}$ | $0.12_{\pm0.20}$ | $\mathbf{5.04_{\pm1.49}}$ | 1.1 |
| | hammer | $0.32_{\pm0.14}$ | $0.51_{\pm0.23}$ | $0.41_{\pm0.08}$ | $0.65_{\pm0.45}$ | $0.24_{\pm0.04}$ | $0.53_{\pm0.35}$ | $\mathbf{19.89_{\pm11.67}}$ | 6.7 |
| human | pen | $8.84_{\pm1.64}$ | $34.43_{\pm2.63}$ | $50.51_{\pm3.43}$ | $13.65_{\pm7.11}$ | $7.36_{\pm8.77}$ | $70.83_{\pm9.13}$ | $\mathbf{77.59_{\pm2.25}}$ | 103.5 |
| | door | $0.02_{\pm0.07}$ | $0.44_{\pm0.43}$ | $1.12_{\pm0.73}$ | $0.41_{\pm0.35}$ | $-0.28_{\pm0.06}$ | $2.87_{\pm0.92}$ | $\mathbf{7.56_{\pm2.30}}$ | 9.9 |
| | hammer | $0.32_{\pm0.14}$ | $0.81_{\pm0.37}$ | $1.21_{\pm0.78}$ | $0.69_{\pm0.09}$ | $0.57_{\pm1.08}$ | $\mathbf{4.68_{\pm0.87}}$ | $5.22_{\pm1.02}$ | 4.4 |
| Average | | 9.38 | 24.63 | 27.28 | 10.26 | 21.58 | 28.16 | **38.45** | 52.22 |

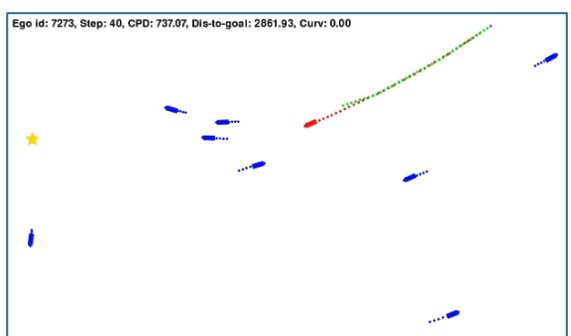

*Figure 1.* Illustration of an episode from the maritime simulation environment. Vessels shown in blue (log-play agents) move along their historical trajectories, the red vessel is controlled by an IL policy with its destination marked by ⋆, and green dots indicate the ego vessel's historical trajectory in the dataset.

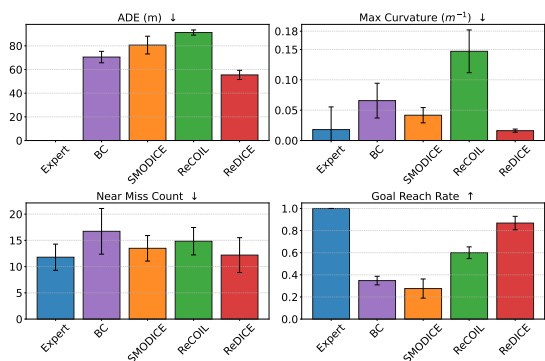

*Figure 2.* Maritime navigation results in ShipNaviSim. Mean and standard deviation are over 5 random seeds.

## 5.3. Application to Maritime Navigation

We evaluate ReDICE on a real-world maritime navigation task where the objective is to imitate how professional pilots maneuver large ships (primarily tankers and cargo vessels) in dense traffic. Beyond serving as a realistic benchmark, learned vessel-behavior models can support safety-critical operations: (i) they can be plugged into a VTIS (Vessel Traffic Information System) pipeline to forecast short-term motions of ships in congestion hotspots such as the Singapore Strait, and (ii) they enable counterfactual stress tests (e.g., simulating a temporary increase in arrivals and assigning learned policies to the injected traffic). Existing RL-based approaches to maritime traffic management are commonly trained online, which demands extensive simulator interaction and high-fidelity vessel dynamics (Singh et al., 2020). Our setting instead focuses on learning directly from offline navigation logs.

We use roughly two years of vessel trajectories in a high-traffic area recorded by the Automatic Identification System (AIS). We instantiate the learning environment using **ShipNaviSim** (Pham et al., 2025), where a single ego vessel is controlled by the learned IL policy and other nearby vessels are replayed from logs (log-play agents), yielding realistic multi-ship encounters (Figure 1). The goal of ego vessel is reaching the destination while avoiding collision with other vessels, and its observation includes a short history of its own states along with the states of nearby vessels. This makes the setting particularly challenging due to the high-dimensional observation space (200 dimensions in our configuration). Since AIS logs provide state sequences but not control inputs, we recover an action representation from consecutive states using the same kinematic inverse model as ShipNaviSim, which allows us to train offline IL methods. We treat AIS trajectories as expert demonstrations, and construct suboptimal data by adding random noise into expert dataset. A comprehensive description of the environment and dataset generation is presented in the Appendix C.2.

Figure 2 shows the comparision results for this setting between ReDICE and other methods. Following the evaluation protocol in ShipNaviSim, we report: **ADE** (Average Displacement Error measuring the difference between trajectoires generated by expert and IL policy; lower is better;), **goal reach rate** (higher is better), **near-miss count** (lower is better; counts close-quarter events during an episode), and kinematic realism via **max curvature** (smaller values indicate less abrupt turning and more physically plausible maneuvers). All results are averaged over five random seeds, with error bars indicating standard deviation. Compared to the expert, all baseline methods including BC, SMODICE, and ReCOIL exhibit substantially lower goal reach rates (≈60%), higher curvature, and increased near-miss counts, indicating difficulties in producing smooth, safe, and reliable vessel behaviors. These shortcomings highlight the challenges of learning from offline data in high-dimensional maritime environment. In contrast, ReDICE consistently improves across all metrics, producing smoother trajectories, fewer near-miss events, and a much higher goal reach rate (≈85%), substantially narrowing the gap to the expert. This shows that ReDICE is effective not only on standard benchmarks but also for real-world applications such as maritime navigation

### 5.4. Ablation Study

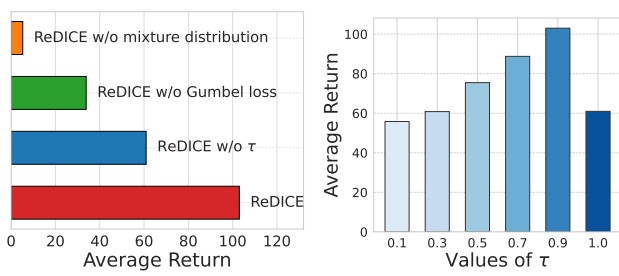

*Figure 3.* Ablation results on different components (**Left**), and $\tau$ values **(Right)**

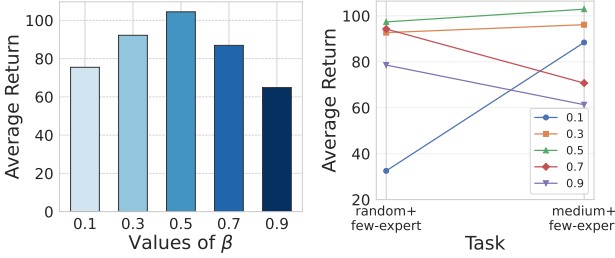

*Figure 4.* Ablation results on different $\beta$

We address **(Q4)** through an ablation study of ReDICE's key components and hyperparameters $\tau$ and $\beta$. Results are shown in Figures 3 and 4, with additional details in Appendix D.

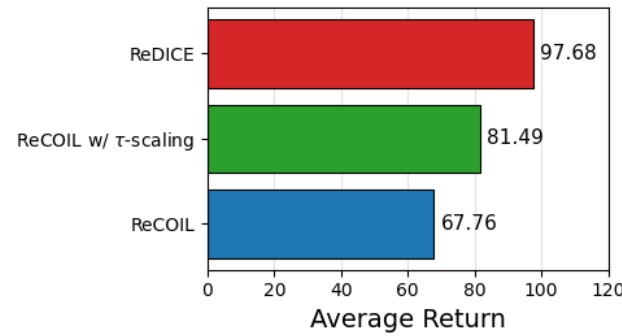

*Figure 5.* Effect of $\tau$-scaling on ReCOIL compared with ReDICE. Results are averaged over 5 random seeds across all D4RL suboptimal tasks.

**Analysis of proposed components:** Figure 3 (left) compares the average return across tasks achieved by ReDICE and several ablated variants, including removing the mixture distribution, the Gumbel-based loss, and the temperature parameter $\tau$. We observe a significant performance degradation when any of these components is removed, indicating that each plays a critical role in the overall effectiveness of ReDICE. In particular, removing the mixture distribution leads to the largest drop (about 20 times) in performance, highlighting the importance of balancing expert and suboptimal data for stable learning.

**Analysis of $\tau$:** According to Figure 3 (right), increasing $\tau$ value initially leads to improved performance by allowing the policy to place greater emphasis on high-advantage actions. However, the performance peaks at $\tau = 0.9$, after which further increases yield diminishing or negative returns. This drop is likely due to overly aggressive advantage weighting, which makes the policy extraction sensitive to errors in the value function. This indicates that $\tau = 0.9$ strikes a favorable balance between stable optimization and effective policy improvement, which is the reason we choose this $\tau$ value for most tasks. In practice, we do not tune $\tau$ over a large range. Following the common practice of tuning policy temperatures in offline methods, we use a small candidate set $\tau \in \{0.9, 1.0\}$. We use $\tau = 0.9$ for most tasks and $\tau = 1.0$ only for a few cases such as Kitchen and human+expert manipulation tasks. Thus, the proposed policy extraction introduces only minimal additional tuning cost.

**Analysis of $\tau$-scaling in ReCOIL:** To further disentangle whether the performance gain comes from the proposed objective or from $\tau$-scaled policy extraction alone, we additionally apply our $\tau$-scaled policy extraction to ReCOIL. Results in Figure 5 show that $\tau$-scaling consistently improves ReCOIL, increasing its average return from 67.76 to 81.49, corresponding to an approximately 20% relative

gain. This demonstrates that our policy extraction mechanism is not specific to ReDICE and can also improve other DICE-based offline IL objectives. However, ReCOIL with $\tau$-scaling still underperforms ReDICE on most tasks, with an average return gap of approximately $20\%$ compared to ReDICE (81.49 vs. 97.68). This indicates that the improvements of ReDICE mainly come from the core objective design rather than from $\tau$-scaling alone.

**Analysis of $\beta$:**   In Figure 4, we study the effect of varying the mixture coefficient $\beta$. We use $\beta = 0.5$ as the default value in all main experiments, following common practice in prior mixture-based IL works such as ReCOIL and related methods. Importantly, this default is not selected based on the ablation results; instead, the ablation is intended to evaluate the robustness of this fixed design choice.

Results show that $\beta = 0.5$ achieves the best average return across tasks, suggesting that equally balancing expert and suboptimal data provides a robust default. Extreme values such as $\beta = 0.1$ or $\beta = 0.9$ can bias learning too heavily toward either suboptimal or expert data, leading to lower average performance. The right panel of Figure 4 further indicates that the effect of $\beta$ depends on the quality and coverage of the supplementary data. For larger $\beta$ values (e.g., 0.7 or 0.9), performance under `random+few-expert`, which has lower expert coverage, is higher than under `medium+few-expert`, whereas the opposite trend holds for smaller $\beta$ values (e.g., 0.1 or 0.3). This suggests that larger $\beta$ values can be more beneficial when expert coverage is limited, while smaller $\beta$ values better exploit informative suboptimal data when coverage is higher.

Overall, $\beta = 0.5$ provides a strong default across settings without requiring prior knowledge of supplementary data quality. Nevertheless, as in many offline IL/RL methods, a fixed hyperparameter may be suboptimal in some outlier cases. When additional validation is available and higher performance is required, we recommend optionally tuning $\beta$ over a small candidate set $\{0.3, 0.5, 0.7\}$.

## 6. Conclusion

We revisited distribution correction estimation for offline imitation learning with supplementary suboptimal data and introduced ReDICE, a stable DICE-based approach derived from a KL-based objective under a mixture distribution. By using the mixture distribution as a reference measure, ReDICE relaxes the coverage assumption required by prior DICE-based offline IL methods while preserving the original expert-imitation objective. We further derive an equivalent dual formulation that reduces to a Gumbel regression objective, improving numerical stability, and propose a $\tau$-scaled policy extraction mechanism that improves performance in practice. Empirically, ReDICE achieves state-of-

the-art performance across standard offline IL benchmarks, remains robust under limited expert coverage, and applies effectively to a real-world maritime navigation setting.

**Limitations and Future Work.**   ReDICE has several limitations that open directions for future work. First, as with prior DICE-based methods, the practical objective relies on an empirical Bellman operator, which is exact under deterministic transitions but may introduce approximation error in highly stochastic environments. Second, ReDICE assumes that expert and supplementary data are collected from the same MDP; extending the method to cross-domain settings with different dynamics may require additional representation alignment or domain adaptation. Third, although we use $\beta = 0.5$ as a robust default following common practice, some outlier tasks may benefit from limited tuning of the mixture coefficient. Finally, ReDICE uses a discriminator to estimate density ratios, and more robust density-ratio estimation methods may further improve performance in high-dimensional or extremely low-data regimes. Future work will study these directions, including stochastic environments, adaptive mixture selection, and larger-scale vision-based offline IL benchmarks such as V-D4RL (Lu et al., 2023).

## Acknowledgements

This work is supported by the Lee Kong Chian Fellowship awarded to Tien Mai.

## Impact Statement

This paper presents work whose goal is to advance the field of Machine Learning, specifically offline imitation learning. There are many potential societal consequences of our work, none of which we feel must be specifically highlighted here.

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

# A. Missing Proofs and Derivations

## A.1. Complete derivation of transforming objective function (3) to (5)

We recall that the primal DICE formulation is as follows:

$$\max_{d \geq 0} \mathbb{E}_{s,a \sim d}[r(s,a)] - \alpha D_f(d(s,a)||d^D(s,a))$$

$$\text{s.t.} \quad \sum_a d(s,a) = (1-\gamma) d_0(s) + \gamma \mathcal{T}_\star d(s), \quad \forall s$$

where $\mathcal{T}_\star d(s) = \sum_{\overline{s},\overline{a}} d(\overline{s},\overline{a}) \mathcal{P}(s \mid \overline{s},\overline{a})$ We apply Lagrangian duality to convert the primal formulation into an unconstrained problem with dual variables $V$:

$$\min_{V(s)} \max_{d \geq 0} \mathbb{E}_{(s,a) \sim d}\Big[r(s,a) - \alpha D_f\big(d(s,a) \| d^D(s,a)\big)\Big] + \sum_s V(s)\Big((1-\gamma)d_0(s) + \gamma \mathcal{T}_\star d(s) - \sum_a d(s,a)\Big)$$

$$= \min_{V(s)} \max_{d \geq 0}(1-\gamma) \sum_s d_0(s) V(s) + \mathbb{E}_{(s,a) \sim d}[r(s,a)] + \sum_{s,a} d(s,a) \sum_{s'} V(s') \mathcal{P}(s' \mid s,a)$$

$$- \sum_{s,a} d(s,a) V(s) - \alpha D_f(d(s,a) \| d^D(s,a)) \tag{25}$$

$$= \min_{V(s)} \max_{w \geq 0}(1-\gamma) \mathbb{E}_{d_0(s)}[V(s)] + \mathbb{E}_{s,a \sim d^D}\left[w(s,a)\Big(r(s,a) + \gamma \sum_{s'} \mathcal{P}(s' \mid s,a) V(s') - V(s)\Big)\right]$$

$$- \mathbb{E}_{s,a \sim d^D}[\alpha f(w(s,a))]. \tag{26}$$

where $w(s,a) = \frac{d(s,a)}{d^D(s,a)}$ is often known as a distribution correction or density ratio. We first focus on solving the inner maximization of (26). Let's denote $\delta(s,a,V) = r(s,a) + \gamma \sum_{s'} \mathcal{P}(s' \mid s,a) V(s') - V(s)$. We can relax the constraint $w \geq 0$ via duality:

$$\min_{\lambda \geq 0} \max_{w(s,a)} \mathbb{E}_{s,a \sim d^D}\big[w(s,a)\delta_V(s,a)\big] - \mathbb{E}_{s,a \sim d^D}\big[\alpha f(w(s,a))\big] + \sum_{s,a} \lambda(s,a) w(s,a). \tag{27}$$

in which $\lambda(s,a)$ are the Lagrange multipliers. Since strong duality holds, we can solve the following KKT conditions:

$$w^*(s,a) \geq 0 \quad \forall s,a \quad \text{(primal feasibility)} \tag{28}$$

$$\lambda^*(s,a) \geq 0 \quad \forall s,a \quad \text{(dual feasibility)} \tag{29}$$

$$d^D(s,a)[-\alpha f'(w^*(s,a)) + \delta(s,a,V) + \lambda^*(s,a)] = 0 <=> \alpha f'(w(s,a)) = \delta(s,a,V) + \lambda^*(s,a) \quad \forall s,a \quad \text{(stationary)} \tag{30}$$

$$\lambda^*(s,a)w^*(s,a) = 0 \quad \forall s,a \quad \text{(complementary slackness)} \tag{31}$$

Applying these conditions, we can derive a closed-form solution of the distribution correction $w^*(s,a) = \max\{0, (f')^{-1}(\frac{\delta(s,a,V)}{\alpha})\}$. This allows us to omit $w$ in (26) to obtain an optimization problem depedning only on $V$:

$$\min_V (1-\gamma)\mathbb{E}_{s \sim d_0}[V(s)] + \alpha \mathbb{E}_{s,a \sim d^D}[f^*(\frac{\delta(s,a,V)}{\alpha})] \tag{32}$$

where $f^*(x) = \max\{0, (f')^{-1}(x)\} \cdot x - f\big(\max\{0, (f')^{-1}(x)\}\big)$ is a variant of convex conjugate.

Let's denote $\mathcal{T}V(s,a) = r(s,a) + \gamma \mathbb{E}_{s' \sim \mathcal{P}(.|s,a)}[V(s')]$ as a Bellman operator on $V$. $\delta(s,a,V)$ is equal to $\mathcal{T}V(s,a) - V(s)$ The objective (32) becomes:

$$\min_V (1-\gamma)\mathbb{E}_{s \sim d_0}[V(s)] + \alpha \mathbb{E}_{s,a \sim d^D}[f^*(\frac{\mathcal{T}V(s,a) - V(s)}{\alpha})] \tag{33}$$

## A.2. Proof of Proposition 4.1

**Corollary A.1.1 (Garg et al., 2021)** *For any valid occupancy measure $\mu$ over state-action pairs and any value function $V$, it holds that*

$$\mathbb{E}_{(s,a)\sim\mu}\big[V(s) - \gamma\,\mathbb{E}_{s'\sim\mathcal{P}(\cdot|s,a)}[V(s')]\big] = (1-\gamma)\,\mathbb{E}_{s\sim p_0}[V(s)]. \tag{34}$$

We use this Corollary to prove the proposition as follows:

*Proof.* We can derive the following result from (34):

$$(1-\gamma)\,\mathbb{E}_{s\sim p_0}[V(s)] = -\mathbb{E}_{(s,a)\sim\mu}[\mathcal{T}V(s,a) - V(s) - r(s,a)] \tag{35}$$

where $\mathcal{T}V(s,a) = r(s,a) + \gamma\mathbb{E}_{s'\sim\mathcal{P}(.|s,a)}[V(s')]$.

Applying this result into the (20) with $f^*(x) = e^x - 1$ we have:

$$
\begin{aligned}
&(1-\gamma)\mathbb{E}_{s\sim d_0}[V(s)] + \mathbb{E}_{s,a\sim d^{mix}}[f^*(\mathcal{T}V(s,a) - V(s))] \\
&= (1-\gamma)\big[\beta\mathbb{E}_{s\sim d_0}[V(s)] + (1-\beta)\mathbb{E}_{s\sim d_0}[V(s)]\big] + \mathbb{E}_{s,a\sim d^{mix}}\Big[e^{\mathcal{T}V(s,a)-V(s)} - 1\Big] \\
&= -\beta\mathbb{E}_{(s,a)\sim d^E}[\mathcal{T}V(s,a) - V(s) - r(s,a)] - (1-\beta)\mathbb{E}_{(s,a)\sim d^S}[\mathcal{T}V(s,a) - V(s) - r(s,a)] \\
&\quad + \mathbb{E}_{d^{mix}}\Big[e^{\mathcal{T}V(s,a)-V(s)}\Big] - 1 \\
&= \mathbb{E}_{d^{mix}}\Big[e^{\mathcal{T}V(s,a)-V(s)}\Big] - \mathbb{E}_{(s,a)\sim d^{mix}}[\mathcal{T}V(s,a) - V(s) - r(s,a)] - 1 \\
&= \mathbb{E}_{d^{mix}}\Big[e^{\mathcal{T}V(s,a)-V(s)}\Big] - \mathbb{E}_{(s,a)\sim d^{mix}}[\mathcal{T}V(s,a) - V(s)] - 1 + \mathbb{E}_{(s,a)\sim d^{mix}}[r(s,a)]
\end{aligned}
$$

Because $r(s,a)$ is a fixed term inferred from a pretrained discriminator, and $d^{mix}$ is a mixture of occupancy measures of fixed datasets, optimizing (20) is equivalent to the following Gumbel regression loss (Garg et al., 2023):

$$\min_V \mathbb{E}_{d^{mix}}\Big[e^{\mathcal{T}V(s,a)-V(s)}\Big] - \mathbb{E}_{(s,a)\sim d^{mix}}[\mathcal{T}V(s,a) - V(s)] - 1$$

$\square$

## A.3. Proof of Proposition 4.2

**Proposition.** *Let's define two nonnegative weights*

$$w_1(s,a) := \exp\big(\mathcal{T}V(s,a) - V(s)\big), \qquad w_\tau(s,a) := \exp\big(\mathcal{T}V(s,a) - \tau V(s)\big),$$

*where $\tau \in [0,1]$. Consider weighted behavior cloning on a fixed dataset distribution $d_{\mathrm{mix}}$:*

$$\max_\pi\ \mathbb{E}_{(s,a)\sim d_{\mathrm{mix}}}\big[w(s,a)\log\pi(a\mid s)\big].$$

*Then the optimal policy for $w = w_1$ is also optimal for $w = w_\tau$ (and vice versa). In particular, both objectives share the same optimal solution $\pi^\star(\cdot\mid s)$.*

*Proof.* Let $\mu(a\mid s) := d_{\mathrm{mix}}(s,a)/d_{\mathrm{mix}}(s)$ be the empirical conditional action distribution of the dataset at state $s$. Since $d_{\mathrm{mix}}(s)$ does not depend on $\pi$, the objective decomposes over states, and for each fixed $s$ we solve

$$\max_{\pi(\cdot|s)\in\Delta(\mathcal{A})}\ \sum_a \mu(a\mid s)\,w(s,a)\,\log\pi(a\mid s). \tag{36}$$

The maximizer of (36) is given by a normalized reweighting:

$$\pi_w^\star(a\mid s) = \frac{\mu(a\mid s)\,w(s,a)}{\sum_b \mu(b\mid s)\,w(s,b)}. \tag{37}$$

Now note that $w_\tau$ differs from $w_1$ by a factor depending only on $s$:

$$w_\tau(s,a) = \exp\big(\mathcal{T}V(s,a) - \tau V(s)\big) = \exp\big((1-\tau)V(s)\big)\exp\big(\mathcal{T}V(s,a) - V(s)\big) = c_\tau(s)\, w_1(s,a),$$

where $c_\tau(s) := \exp((1-\tau)V(s))$ is independent of $a$. Plugging into (37),

$$\pi^\star_{w_\tau}(a \mid s) = \frac{\mu(a \mid s)\, c_\tau(s)\, w_1(s,a)}{\sum_b \mu(b \mid s)\, c_\tau(s)\, w_1(s,b)} = \frac{\mu(a \mid s)\, w_1(s,a)}{\sum_b \mu(b \mid s)\, w_1(s,b)} = \pi^\star_{w_1}(a \mid s).$$

Thus, the two weighted BC objectives have exactly the same optimal solution $\pi^\star(\cdot \mid s)$. $\qquad\qquad\square$

## B. More discussion on difference between ReDICE and ReCOIL

Both ReDICE and ReCOIL (Sikchi et al., 2024) addresse coverage issues via mixture distribution, but ReCOIL's objective differs from ours in an important way. Let $d^{\mathrm{mix}}(s,a) = \beta d^E(s,a) + (1-\beta)d^S(s,a)$ Consider the following three objectives.

**Original IL objective.**

$$\min_{d^\pi} D_f\left(d^\pi(s,a) \,\|\, d^E(s,a)\right). \tag{38}$$

**ReCOIL objective.**

$$\min_{\pi,d} D_f\left(\beta d(s,a) + (1-\beta)d^S(s,a) \,\|\, d^{\mathrm{mix}}(s,a)\right) \tag{39}$$

$$\text{s.t.} \quad d(s,a) = (1-\gamma)d_0(s)\pi(a|s) + \gamma\sum_{s',a'} d(s',a')\mathcal{P}(s|s',a')\pi(a|s).$$

**ReDICE objective.**

$$\min_d \mathbb{E}_{d(s,a)}\left[\log\frac{d^{\mathrm{mix}}(s,a)}{d^E(s,a)}\right] + D_{\mathrm{KL}}\left(d(s,a) \,\|\, d^{\mathrm{mix}}(s,a)\right) \tag{40}$$

$$\text{s.t.} \quad \sum_a d(s,a) = (1-\gamma)d_0(s) + \gamma\sum_{s',a'} d(s',a')\mathcal{P}(s|s',a').$$

The constraints in both (39) and (40) are Bellman flow constraints to ensure that $d(s,a)$ is a valid occupancy measure $d^\pi(s,a)$ in (38). From Eqs. (38)–(40), ReDICE has three key advantages over ReCOIL:

- **Preserves the original objective while relaxing coverage.** ReCOIL matches divergences between mixture distributions that only share the same optimum as (1), whereas ReDICE remains exactly equivalent to (1) under KL.

- **Avoids adversarial max–min optimization.** The dual formulation of ReCOIL solves a max–min problem over $Q$ and $\pi$ (Theorem 1 in (Sikchi et al., 2024)), which may induce extrapolation errors from out-of-distribution actions in offline settings (Kumar et al., 2020). ReDICE reduces to a minimization over $V$, avoiding this issue.

- **Simpler and more theory-aligned optimization.** To mitigate the OOD issues discussed above, ReCOIL relies on an in-sample surrogate objective to jointly learn $Q(s,a)$ and $V(s)$ (Eqs. (11)–(12) in (Sikchi et al., 2024)). This introduces an additional hyperparameter $\tau$ that must be tuned per task (Appendix F.1 of (Sikchi et al., 2024)). As a result, the practical implementation of ReCOIL creates a gap between theory and practice and increases tuning complexity. In contrast, ReDICE uses a single minimization over $V$, reducing tuning complexity.

## C. Experiment Setup and Implementation Details

We implement our approach in PyTorch and run experiments on a multi-GPU cluster consisting of NVIDIA RTX 3090 GPUs. Each run evaluates five random seeds in parallel on shared hardware resources (one GPU, 32 CPU cores, and 128 GB RAM). All methods are trained for one million gradient steps and evaluated every 5,000 steps.

| Environment | State Dim | Action Dim | Suboptimal Dataset | Dataset Details |
|---|---|---|---|---|
| Hopper | 11 | 3 | `random+expert` `medium+expert` `random+few-expert` `medium+few-expert` | 1e6 random transitions + 200 expert trajectories 1e6 medium transitions + 200 expert trajectories 1e6 random transitions + 30 expert trajectories 1e6 medium transitions + 30 expert trajectories |
| Walker2d | 17 | 6 | `random+expert` `medium+expert` `random+few-expert` `medium+few-expert` | 1e6 random transitions + 200 expert trajectories 1e6 medium transitions + 200 expert trajectories 1e6 random transitions + 30 expert trajectories 1e6 medium transitions + 30 expert trajectories |
| Halfcheetah | 17 | 6 | `random+expert` `medium+expert` `random+few-expert` `medium+few-expert` | 1e6 random transitions + 200 expert trajectories 1e6 medium transitions + 200 expert trajectories 1e6 random transitions + 30 expert trajectories 1e6 medium transitions + 30 expert trajectories |
| Ant | 27 | 8 | `random+expert` `medium+expert` `random+few-expert` `medium+few-expert` | 1e6 random transitions + 200 expert trajectories 1e6 medium transitions + 200 expert trajectories 1e6 random transitions + 30 expert trajectories 1e6 medium transitions + 30 expert trajectories |
| Pen | 45 | 24 | `cloned+expert` `human+expert` | 5e6 cloned transitions + 30 expert trajectories 5000 human transitions + 30 expert trajectories |
| Door | 39 | 28 | `cloned+expert` `human+expert` | 1e6 cloned transitions + 30 expert trajectories 6729 human transitions + 30 expert trajectories |
| Hammer | 46 | 26 | `cloned+expert` `human+expert` | 1e6 cloned transitions + 30 expert trajectories 11310 human transitions + 30 expert trajectories |
| Kitchen | 59 | 9 | `partial+expert` `mixed+expert` | 136950 partial transitions + 1 expert trajectory 136950 mixed transitions + 1 expert trajectory |

*Table 3.* Overview of D4RL tasks and their respective suboptimal datasets used in the offline IL setting.

### C.1. D4RL tasks

We adopt the same offline IL benchmark introduced in ReCOIL (Sikchi et al., 2024), which is constructed using datasets from the D4RL framework (Fu et al., 2020) and evaluated on MuJoCo control tasks. Across all environments, the expert dataset consists of a single expert trajectory. For locomotion tasks, suboptimal datasets—denoted as `random+expert`, `random+few-expert`, `medium+expert`, and `medium+few-expert` are created by combining expert demonstrations with lower-quality trajectories from the D4RL random-v2 and medium-v2 datasets. Specifically, the `random+expert` and `medium+expert` settings include 200 expert trajectories mixed with approximately one million transitions from the corresponding D4RL dataset, while the `x+few-expert` variants use only 30 expert trajectories. For manipulation tasks, each `x+expert` dataset is formed by mixing 30 expert trajectories with the full corresponding D4RL dataset, and we consistently use the -v0 versions of D4RL for these environments. An overview of all tasks is provided in Table 3.

### C.2. ShipNaviSim task

The Maritime Navigation task was developed using historical data from a high-density traffic hotspot in the Singapore Strait, adhering to the AIS-driven simulation paradigm established in **ShipNaviSim** (Pham et al., 2025). We defined our planning region by selecting an area characterized by peak traffic volume and elevated collision risk, where numerous vessel paths intersect. Two years of AIS data were sourced from MarineTraffic (https://www.marinetraffic.com/), encompassing both static vessel attributes (dimensions, type, and ID) and dynamic movement variables (coordinates, speed, heading, and course-over-ground). Our analysis focuses on tankers and cargo ships, as their significant scale (200–300 meters) and limited maneuverability represent the highest navigational risk. Following a 10-second interpolation process,

the final dataset comprises 125,000 trajectories and 14 million environment transitions, with an average trajectory length of 100–150 steps. The data is divided into an 80% training set and a 20% evaluation set.

**Observation Space**    The observation space is constructed from the view of the ego agent, providing the necessary context for multi-vessel interaction and decision-making. At each time step, the agent receives a historical sequence of its own state and the states of the 10 nearest vessels over a predefined window of past steps. Each historical data point includes the $x$ and $y$ coordinates, speed $v$, and heading $h$. The agent is also provided with its target goal location. To ensure a fair comparison across different algorithms, we utilized a consistent, straightforward neural network architecture. Rather than employing complex structural layers, the observation space (200-dimensional) is flattened and fed directly into the network.

**Action Space**    We define the action space as a 3-dimensional continuous space representing changes in state, denoted as $\langle d_x, d_y, d_h \rangle$. These components correspond to the change in $x$ and $y$ coordinates and the change in heading $h$, respectively. The vessel's velocity for the following step is calculated based on the displacement over a fixed time interval ($\delta_T = 10$ seconds), where $v_{t+1} = \sqrt{d_x^2 + d_y^2}/\delta_T$. This *delta action space* formulation (Gulino et al., 2023) is applicable to various mobile agents. By representing actions as the difference between consecutive states, we can utilize a simple Inverse Kinematics Model to derive the necessary actions from the state-only trajectories present in our historical dataset.

**Evaluation Metrics**    To assess the performance of the navigation policies against human expert behavior, we adopt the *vessel-specific metrics* established in **ShipNaviSim**:

- **Goal-Conditioned ADE (GC-ADE)**: This metric quantifies the average Euclidean distance between the trajectory generated by the learned policy and the ground-truth historical trajectory. For a modeled trajectory $\tau_m$ and a predicted trajectory $\tau_p$, the error is averaged over the duration of the shorter sequence:

$$\text{GC-ADE} = \frac{1}{\min(T_m, T_p)} \sqrt{\sum_{t=1}^{\min(T_m, T_p)} (x_t^m - x_t^p)^2 + (y_t^m - y_t^p)^2}$$

- **Goal Rate**: This represents the success frequency of the ego agent in reaching its destination. A trial is considered successful if the agent comes within a 200-meter radius of the goal.

- **Near Miss Count**: This measures the average number of time-steps per episode where the ego agent is within 3 cable lengths (555 meters) of another vessel. While domain experts categorize this proximity as a "near-miss," we use this metric as a broader indicator of traffic density and potential navigational risk, rather than an absolute measure of imminent collision.

- **Max Curvature**: Curvature measures the angle change in the vessel's course over ground at each time step. It indicates the sharpness of a turn and helps assess the maneuverability and alignment with expected vessel behavior, which cannot make sharp turns. By monitoring maximum curvature, we can ensure that the generated trajectories adhere to the physical motion constraints of real-world maritime vessels and avoid erratic or unrealistic maneuvers.

## C.3. Hyperparameters

Our implementation is mainly based on the official codebases of ODICE (Mao et al., 2024) and SMODICE (Ma et al., 2022), adopting their core network configurations and parameters as detailed in Table 4. To enhance training stability, we follow (Garg et al., 2023), incorporating Layer Normalization (Ba et al., 2016) into the critic network. We set a fixed mixture ratio $\beta = 0.5$ for all experiments For the parameter $\tau$, we find that $\tau = 0.9$ works for most tasks except the Kitchen and `human+expert` tasks where $\tau = 1.0$ yields better results. We also note that other SOTA offline IL methods such as ReCOIL (Sikchi et al., 2024) and SMODICE (Ma et al., 2022), also adjust their hyperparameters for different sets of tasks. We follow such tuned hyperaparameters as reported in the respective papers for different baselines for fair comparisons.

*Table 4.* Hyperparameters of ReDICE

| Type | Hyperparameter | Value |
|---|---|---|
| Discriminator | Network Size | [256, 256] |
| | Activation Function | Tanh |
| | Learning Rate | 3e-4 |
| | Training Length | 5000 |
| | Batch Size | 256 |
| | Optimizer | Adam |
| Actor | Network Size | [1024, 1024] |
| | Activation Function | ReLU |
| | Learning Rate | 3e-4 |
| | Weight Decay | 1e-3 |
| | Training Length | 1M steps |
| | Batch Size | 512 |
| | Optimizer | Adam |
| | Dropout Rate | 0.1 |
| | LR decay schedule | cosine |
| Critic | Network Size | [256, 256] |
| | Activation Function | ReLU |
| | Learning Rate | 3e-4 |
| | Weight Decay | 5e-6 |
| | Training Length | 1M steps |
| | Batch Size | 512 |
| | Optimizer | Adam |
| | Soft Target Update Rate | 0.005 |
| | Discount Factor $\gamma$ | 0.99 |

# D. Detailed Ablation Results

*Table 5.* Average normalized return over last 10 evaluations of ReDICE against its variants on the D4RL suboptimal datasets with 1 expert trajectory. The mean and std are obtained over 5 random seeds. Offline IL methods with avg. perf within the std-dev of the top performing approach is in **bold**.

| Suboptimal Dataset | Env | ReDICE w/o mixture distribution | ReDICE w/o Gumbel loss | ReDICE w/o $\tau$ | ReDICE | Expert |
|---|---|---|---|---|---|---|
| random+ expert | hopper | $0.93_{\pm0.02}$ | $6.79_{\pm2.80}$ | $52.09_{\pm2.81}$ | $\mathbf{109.80}_{\pm0.31}$ | 111.3 |
| | halfcheetah | $2.25_{\pm0.00}$ | $2.24_{\pm0.01}$ | $23.88_{\pm1.65}$ | $\mathbf{91.82}_{\pm0.44}$ | 88.83 |
| | walker2d | $-0.14_{\pm0.03}$ | $3.59_{\pm3.95}$ | $73.97_{\pm14.88}$ | $\mathbf{107.98}_{\pm0.12}$ | 106.92 |
| | ant | $27.41_{\pm0.29}$ | $53.09_{\pm5.52}$ | $82.22_{\pm4.33}$ | $\mathbf{126.32}_{\pm0.81}$ | 130.75 |
| random+ few-expert | hopper | $1.04_{\pm0.13}$ | $2.92_{\pm1.00}$ | $29.58_{\pm0.41}$ | $\mathbf{93.63}_{\pm1.69}$ | 111.33 |
| | halfcheetah | $2.25_{\pm0.00}$ | $2.23_{\pm0.00}$ | $20.12_{\pm1.24}$ | $\mathbf{79.31}_{\pm4.59}$ | 88.83 |
| | walker2d | $-0.17_{\pm0.06}$ | $1.54_{\pm1.39}$ | $7.37_{\pm1.02}$ | $\mathbf{108.26}_{\pm0.03}$ | 106.92 |
| | ant | $30.53_{\pm1.93}$ | $33.24_{\pm1.43}$ | $47.32_{\pm5.95}$ | $\mathbf{108.15}_{\pm3.59}$ | 130.75 |
| medium+ expert | hopper | $0.71_{\pm0.04}$ | $53.85_{\pm0.61}$ | $93.74_{\pm3.28}$ | $\mathbf{111.01}_{\pm0.38}$ | 111.33 |
| | halfcheetah | $2.25_{\pm0.00}$ | $42.74_{\pm0.18}$ | $76.83_{\pm2.79}$ | $\mathbf{91.80}_{\pm0.42}$ | 88.83 |
| | walker2d | $-0.21_{\pm0.08}$ | $72.75_{\pm1.27}$ | $\mathbf{108.90}_{\pm0.20}$ | $\mathbf{108.40}_{\pm0.50}$ | 106.92 |
| | ant | $1.20_{\pm12.79}$ | $93.43_{\pm2.58}$ | $116.03_{\pm6.37}$ | $\mathbf{123.24}_{\pm1.48}$ | 130.75 |
| medium few-expert | hopper | $9.14_{\pm14.53}$ | $54.06_{\pm0.47}$ | $38.41_{\pm6.53}$ | $\mathbf{109.50}_{\pm1.23}$ | 111.33 |
| | halfcheetah | $2.25_{\pm0.00}$ | $42.39_{\pm0.14}$ | $53.44_{\pm1.02}$ | $\mathbf{81.57}_{\pm2.94}$ | 88.83 |
| | walker2d | $0.00_{\pm0.25}$ | $69.03_{\pm2.36}$ | $75.25_{\pm16.88}$ | $\mathbf{108.70}_{\pm0.44}$ | 106.92 |
| | ant | $-11.24_{\pm9.95}$ | $88.57_{\pm3.51}$ | $\mathbf{109.66}_{\pm4.20}$ | $\mathbf{111.73}_{\pm2.37}$ | 130.75 |
| cloned+expert | pen | $31.21_{\pm3.61}$ | $20.57_{\pm16.71}$ | $61.78_{\pm10.00}$ | $\mathbf{88.84}_{\pm8.84}$ | 106.42 |
| | door | $-0.00_{\pm0.01}$ | $0.38_{\pm0.66}$ | $44.29_{\pm35.07}$ | $\mathbf{92.23}_{\pm2.45}$ | 103.94 |
| | hammer | $0.31_{\pm0.01}$ | $0.29_{\pm0.01}$ | $43.19_{\pm16.88}$ | $\mathbf{104.86}_{\pm10.07}$ | 125.71 |
| Average | | 5.25 | 33.88 | 60.95 | **103.01** | 109.86 |

*Table 6.* Average normalized return over last 10 evaluations of ReDICE with different $\tau$ values on the D4RL suboptimal datasets with 1 expert trajectory. The mean and std are obtained over 5 random seeds. Offline IL methods with avg. perf within the std-dev of the top performing approach is in **bold**.

| Suboptimal Dataset | Env | $\tau=1.0$ | $\tau=0.9$ | $\tau=0.7$ | $\tau=0.5$ | $\tau=0.3$ | $\tau=0.1$ | Expert |
|---|---|---|---|---|---|---|---|---|
| random+ expert | hopper | $52.09_{\pm2.81}$ | $\mathbf{109.80}_{\pm0.31}$ | $109.75_{\pm1.07}$ | $26.95_{\pm12.75}$ | $10.64_{\pm2.84}$ | $12.17_{\pm4.30}$ | 111.3 |
| | halfcheetah | $23.88_{\pm1.65}$ | $\mathbf{91.82}_{\pm0.44}$ | $92.02_{\pm0.05}$ | $91.72_{\pm0.13}$ | $91.32_{\pm0.75}$ | $90.84_{\pm0.79}$ | 88.83 |
| | walker2d | $73.97_{\pm14.88}$ | $\mathbf{107.98}_{\pm0.12}$ | $108.25_{\pm0.19}$ | $103.57_{\pm7.78}$ | $49.26_{\pm16.42}$ | $22.45_{\pm19.59}$ | 106.92 |
| | ant | $82.22_{\pm4.33}$ | $\mathbf{126.32}_{\pm0.81}$ | $114.33_{\pm2.29}$ | $78.93_{\pm8.83}$ | $65.70_{\pm4.73}$ | $59.54_{\pm3.46}$ | 130.75 |
| random+ few-expert | hopper | $29.58_{\pm0.41}$ | $\mathbf{93.63}_{\pm1.69}$ | $6.72_{\pm1.03}$ | $5.95_{\pm0.71}$ | $7.02_{\pm0.98}$ | $6.43_{\pm0.94}$ | 111.33 |
| | halfcheetah | $20.12_{\pm1.24}$ | $\mathbf{79.31}_{\pm4.59}$ | $74.74_{\pm2.26}$ | $69.62_{\pm6.59}$ | $64.01_{\pm7.41}$ | $61.79_{\pm9.49}$ | 88.83 |
| | walker2d | $7.37_{\pm1.02}$ | $\mathbf{108.26}_{\pm0.03}$ | $108.09_{\pm0.61}$ | $53.47_{\pm8.50}$ | $5.99_{\pm3.30}$ | $4.66_{\pm2.80}$ | 106.92 |
| | ant | $47.32_{\pm5.95}$ | $\mathbf{108.15}_{\pm3.59}$ | $58.83_{\pm6.68}$ | $29.16_{\pm1.18}$ | $27.89_{\pm2.43}$ | $27.59_{\pm1.58}$ | 130.75 |
| medium+ expert | hopper | $93.74_{\pm3.28}$ | $\mathbf{111.01}_{\pm0.38}$ | $110.02_{\pm0.29}$ | $110.55_{\pm0.03}$ | $110.24_{\pm0.23}$ | $109.81_{\pm0.94}$ | 111.33 |
| | halfcheetah | $76.83_{\pm2.79}$ | $\mathbf{91.80}_{\pm0.42}$ | $89.98_{\pm0.29}$ | $55.92_{\pm1.50}$ | $43.79_{\pm0.38}$ | $42.60_{\pm0.20}$ | 88.83 |
| | walker2d | $\mathbf{108.90}_{\pm0.20}$ | $\mathbf{108.40}_{\pm0.50}$ | $108.06_{\pm0.02}$ | $108.15_{\pm0.13}$ | $107.79_{\pm0.76}$ | $107.42_{\pm0.41}$ | 106.92 |
| | ant | $116.03_{\pm6.37}$ | $\mathbf{123.24}_{\pm1.48}$ | $108.85_{\pm4.98}$ | $98.13_{\pm4.33}$ | $93.97_{\pm4.44}$ | $94.17_{\pm2.93}$ | 130.75 |
| medium few-expert | hopper | $38.41_{\pm6.53}$ | $\mathbf{109.50}_{\pm1.23}$ | $109.87_{\pm1.37}$ | $108.69_{\pm0.57}$ | $102.66_{\pm7.28}$ | $107.36_{\pm2.39}$ | 111.33 |
| | halfcheetah | $53.44_{\pm1.02}$ | $\mathbf{81.57}_{\pm2.94}$ | $46.10_{\pm1.57}$ | $43.53_{\pm0.65}$ | $42.67_{\pm0.35}$ | $42.61_{\pm0.29}$ | 88.83 |
| | walker2d | $75.25_{\pm16.88}$ | $\mathbf{108.70}_{\pm0.44}$ | $106.74_{\pm0.80}$ | $106.27_{\pm0.03}$ | $104.21_{\pm1.92}$ | $58.77_{\pm6.18}$ | 106.92 |
| | ant | $\mathbf{109.66}_{\pm4.20}$ | $\mathbf{111.73}_{\pm2.37}$ | $93.30_{\pm0.41}$ | $92.07_{\pm3.43}$ | $87.74_{\pm4.17}$ | $84.99_{\pm2.74}$ | 130.75 |
| cloned+expert | pen | $61.78_{\pm10.00}$ | $\mathbf{88.84}_{\pm8.84}$ | $66.91_{\pm6.94}$ | $73.45_{\pm7.47}$ | $67.32_{\pm5.54}$ | $59.04_{\pm6.99}$ | 106.42 |
| | door | $44.29_{\pm35.07}$ | $\mathbf{92.23}_{\pm2.45}$ | $89.98_{\pm18.85}$ | $91.94_{\pm10.27}$ | $27.67_{\pm47.87}$ | $25.15_{\pm43.65}$ | 103.94 |
| | hammer | $43.19_{\pm16.88}$ | $\mathbf{104.86}_{\pm10.07}$ | $83.83_{\pm36.16}$ | $85.82_{\pm37.01}$ | $45.77_{\pm50.51}$ | $42.99_{\pm56.75}$ | 125.71 |
| Average | | 60.95 | **103.01** | 88.76 | 75.47 | 60.82 | 55.81 | 109.86 |

*Table 7.* Average normalized return over last 10 evaluations of ReDICE with different $\beta$ values on the D4RL suboptimal datasets with 1 expert trajectory. The mean and std are obtained over 5 random seeds. Offline IL methods with avg. perf within the std-dev of the top performing approach is in **bold**.

| Suboptimal Dataset | Env | $\beta = 0.1$ | $\beta = 0.3$ | $\beta = 0.5$ | $\beta = 0.7$ | $\beta = 0.9$ | Expert |
|---|---|---|---|---|---|---|---|
| random+ 
 expert | hopper 
 halfcheetah 
 walker2d 
 ant | $105.94_{\pm4.96}$ 
 $89.26_{\pm0.09}$ 
 $\mathbf{108.13_{\pm0.23}}$ 
 $69.52_{\pm4.44}$ | $\mathbf{109.97_{\pm0.49}}$ 
 $91.20_{\pm0.37}$ 
 $\mathbf{108.19_{\pm0.17}}$ 
 $68.28_{\pm9.99}$ | $\mathbf{109.80_{\pm0.31}}$ 
 $\mathbf{91.82_{\pm0.44}}$ 
 $\mathbf{107.98_{\pm0.12}}$ 
 $\mathbf{126.32_{\pm0.81}}$ | $102.21_{\pm2.53}$ 
 $91.26_{\pm0.50}$ 
 $107.96_{\pm0.50}$ 
 $102.60_{\pm18.49}$ | $65.84_{\pm1.69}$ 
 $81.60_{\pm0.20}$ 
 $86.77_{\pm3.08}$ 
 $120.33_{\pm4.19}$ | $111.3$ 
 $88.83$ 
 $106.92$ 
 $130.75$ |
| random+ 
 few-expert | hopper 
 halfcheetah 
 walker2d 
 ant | $6.20_{\pm0.55}$ 
 $2.32_{\pm0.04}$ 
 $2.68_{\pm0.27}$ 
 $\mathbf{118.82_{\pm1.57}}$ | $88.31_{\pm3.62}$ 
 $56.46_{\pm3.33}$ 
 $107.75_{\pm0.68}$ 
 $\mathbf{118.19_{\pm1.88}}$ | $\mathbf{93.63_{\pm1.69}}$ 
 $79.31_{\pm4.59}$ 
 $\mathbf{108.26_{\pm0.03}}$ 
 $108.15_{\pm3.59}$ | $80.43_{\pm6.40}$ 
 $\mathbf{82.75_{\pm1.17}}$ 
 $\mathbf{108.31_{\pm0.20}}$ 
 $105.52_{\pm16.85}$ | $75.28_{\pm19.26}$ 
 $59.03_{\pm26.80}$ 
 $88.64_{\pm20.27}$ 
 $91.34_{\pm3.23}$ | $111.33$ 
 $88.83$ 
 $106.92$ 
 $130.75$ |
| medium+ 
 expert | hopper 
 halfcheetah 
 walker2d 
 ant | $110.00_{\pm1.67}$ 
 $42.45_{\pm0.19}$ 
 $107.48_{\pm0.69}$ 
 $90.88_{\pm2.59}$ | $\mathbf{110.67_{\pm0.50}}$ 
 $42.51_{\pm0.07}$ 
 $108.08_{\pm0.05}$ 
 $89.27_{\pm1.31}$ | $\mathbf{111.01_{\pm0.38}}$ 
 $\mathbf{91.80_{\pm0.42}}$ 
 $\mathbf{108.40_{\pm0.50}}$ 
 $\mathbf{123.24_{\pm1.48}}$ | $60.00_{\pm46.69}$ 
 $63.33_{\pm16.82}$ 
 $\mathbf{108.37_{\pm0.11}}$ 
 $94.92_{\pm2.96}$ | $32.35_{\pm8.38}$ 
 $7.12_{\pm2.01}$ 
 $8.32_{\pm1.65}$ 
 $75.76_{\pm3.49}$ | $111.33$ 
 $88.83$ 
 $106.92$ 
 $130.75$ |
| medium 
 few-expert | hopper 
 halfcheetah 
 walker2d 
 ant | $102.84_{\pm5.21}$ 
 $81.17_{\pm6.46}$ 
 $80.84_{\pm2.58}$ 
 $88.74_{\pm3.67}$ | $\mathbf{108.49_{\pm1.49}}$ 
 $\mathbf{83.06_{\pm2.95}}$ 
 $104.84_{\pm2.61}$ 
 $88.11_{\pm0.65}$ | $\mathbf{109.50_{\pm1.23}}$ 
 $\mathbf{81.57_{\pm2.94}}$ 
 $\mathbf{108.70_{\pm0.44}}$ 
 $\mathbf{111.73_{\pm2.37}}$ | $27.07_{\pm14.61}$ 
 $\mathbf{85.37_{\pm3.87}}$ 
 $81.81_{\pm22.97}$ 
 $88.58_{\pm2.47}$ | $15.00_{\pm2.60}$ 
 $58.88_{\pm5.56}$ 
 $80.22_{\pm20.55}$ 
 $91.10_{\pm4.46}$ | $111.33$ 
 $88.83$ 
 $106.92$ 
 $130.75$ |
| Average | | $75.45$ | $92.71$ | $\mathbf{104.45}$ | $86.91$ | $64.85$ | $109.46$ |

*Table 8.* Performance comparison of SMODICE variants on D4RL suboptimal datasets with 1 expert trajectories. The mean and std are obtained over multiple random seeds. Methods with average performance within one standard deviation of the top-performing approach are shown in **bold**.

| Suboptimal Dataset | Env | SMODICE | SMODICE w/ orthogonal | SMODICE w/ all |
|---|---|---|---|---|
| random+ 
 expert | hopper 
 halfcheetah 
 walker2d 
 ant | $\mathbf{104.09_{\pm1.82}}$ 
 $\mathbf{80.92_{\pm1.94}}$ 
 $\mathbf{107.46_{\pm0.22}}$ 
 $\mathbf{126.48_{\pm1.78}}$ | $5.79_{\pm0.30}$ 
 $2.25_{\pm0.00}$ 
 $1.29_{\pm0.07}$ 
 $38.66_{\pm9.82}$ | $0.93_{\pm0.02}$ 
 $2.25_{\pm0.00}$ 
 $-0.14_{\pm0.03}$ 
 $27.41_{\pm0.29}$ |
| random+ 
 few-expert | hopper 
 halfcheetah 
 walker2d 
 ant | $\mathbf{58.72_{\pm7.09}}$ 
 $\mathbf{4.16_{\pm2.83}}$ 
 $\mathbf{3.48_{\pm2.57}}$ 
 $13.26_{\pm5.01}$ | $5.34_{\pm0.07}$ 
 $2.25_{\pm0.00}$ 
 $1.40_{\pm0.03}$ 
 $\mathbf{29.50_{\pm0.32}}$ | $1.04_{\pm0.13}$ 
 $2.25_{\pm0.00}$ 
 $-0.17_{\pm0.06}$ 
 $\mathbf{30.53_{\pm1.93}}$ |
| medium+ 
 expert | hopper 
 halfcheetah 
 walker2d 
 ant | $\mathbf{57.83_{\pm3.52}}$ 
 $\mathbf{66.52_{\pm2.32}}$ 
 $39.71_{\pm10.78}$ 
 $\mathbf{109.73_{\pm1.88}}$ | $52.89_{\pm0.25}$ 
 $42.67_{\pm0.08}$ 
 $\mathbf{72.71_{\pm1.08}}$ 
 $99.40_{\pm3.99}$ | $0.71_{\pm0.04}$ 
 $2.25_{\pm0.00}$ 
 $-0.21_{\pm0.08}$ 
 $1.20_{\pm12.79}$ |
| medium+ 
 few-expert | hopper 
 halfcheetah 
 walker2d 
 ant | $\mathbf{53.33_{\pm0.80}}$ 
 $\mathbf{41.47_{\pm1.99}}$ 
 $13.78_{\pm14.21}$ 
 $89.22_{\pm2.44}$ | $52.83_{\pm0.98}$ 
 $42.38_{\pm0.29}$ 
 $\mathbf{71.75_{\pm1.62}}$ 
 $89.90_{\pm0.59}$ | $9.14_{\pm14.53}$ 
 $2.25_{\pm0.00}$ 
 $0.00_{\pm0.25}$ 
 $-11.24_{\pm9.95}$ |
| cloned+expert | pen 
 door 
 hammer | $\mathbf{33.97_{\pm19.99}}$ 
 $\mathbf{3.57_{\pm3.38}}$ 
 $\mathbf{3.98_{\pm5.45}}$ | $22.25_{\pm2.44}$ 
 $0.01_{\pm0.00}$ 
 $1.09_{\pm0.56}$ | $\mathbf{31.21_{\pm3.61}}$ 
 $-0.00_{\pm0.01}$ 
 $0.31_{\pm0.01}$ |
| Average | | $\mathbf{53.25}$ | $33.39$ | $5.25$ |

*Table 9.* Performance comparison of ReCOIL variants and ReDICE on D4RL suboptimal datasets with 1 expert trajectory. The reported values are normalized returns averaged over multiple random seeds. The best-performing method for each task is shown in **bold**. We evaluate ReCOIL with $\tau$-scaling by replacing its policy extraction and tuning $\tau \in \{0.5, 0.7, 0.9, 1.0\}$. We observe consistent trends with ReDICE: for ReCOIL, $\tau = 0.9$ also performs best on locomotion and cloned tasks, while $\tau = 1.0$ works better on Kitchen and human datasets.

| Dataset | Env | ReCOIL w/ $\tau$-scaling | ReCOIL | ReDICE |
|---|---|---|---|---|
| random+expert | hopper | 102.79 | 98.33 | **109.80** |
| | halfcheetah | 90.94 | 78.78 | **91.82** |
| | walker2d | 71.79 | 105.72 | **107.98** |
| | ant | 126.07 | 116.43 | **126.32** |
| random+few-expert | hopper | 90.15 | 86.65 | **93.63** |
| | halfcheetah | 37.44 | 5.25 | **79.31** |
| | walker2d | 34.05 | 12.59 | **108.26** |
| | ant | 96.71 | 56.68 | **108.15** |
| medium+expert | hopper | 110.84 | 85.25 | **111.01** |
| | halfcheetah | 75.91 | 78.88 | **91.80** |
| | walker2d | **108.40** | 108.27 | **108.40** |
| | ant | 123.02 | 119.21 | **123.24** |
| medium+few-expert | hopper | 92.59 | 38.88 | **109.50** |
| | halfcheetah | 57.85 | 41.90 | **81.57** |
| | walker2d | 107.86 | 71.44 | **108.70** |
| | ant | 108.79 | 104.03 | **111.73** |
| cloned+expert | pen | 40.80 | 45.59 | **88.84** |
| | door | 87.61 | 16.18 | **92.23** |
| | hammer | 31.65 | 2.60 | **104.86** |
| human+expert | pen | 84.35 | **85.04** | 82.62 |
| | door | 92.36 | 84.02 | **94.31** |
| | hammer | 80.41 | 86.45 | **107.97** |
| partial+expert | kitchen | **54.25** | 48.80 | 54.00 |
| mixed+expert | kitchen | 49.17 | **49.35** | 48.25 |
| Average | | 81.49 | 67.76 | **97.68** |

