# OpenReview forum: "Revisiting Distribution Correction Estimation for Offline Imitation Learning with Suboptimal Dataset"
_ICML.cc/2026/Conference — ICML 2026 regular_

### Official Review · Reviewer_sgma · 2026-03-06

**Soundness:** 3
**Presentation:** 2
**Significance:** 3
**Originality:** 3
**Overall Recommendation:** 3
**Confidence:** 3

**Summary:**

This paper proposes ReDICE to address the limited expert data problem in offline imitation learning. ReDICE decomposes the KL divergence using a mixture of expert and suboptimal data distributions, removing the reliance on the coverage assumption. The authors further show that the dual objective is equivalent to a Gumbel regression loss, which is strictly convex with a unique global optimum. ReDICE outperforms existing baselines on 24 D4RL tasks and a real-world maritime navigation task.

**Compliance With Llm Reviewing Policy:**

Affirmed.

**Final Justification:**

The rebuttal partially addresses my concerns. The discriminator training loss curves provide reasonable evidence of stability, and the β ablation supports β=0.5 as a reasonable default, though the 9.0% gap on ant+random+few-expert remains non-trivial. However, the pipeline figure provided is informal and does not meet publication standards. The incremental nature of the contribution relative to SMODICE and ReCOIL remains a concern. I maintain my original score.

**Key Questions For Authors:**

1. The high variance on door and hammer tasks makes me wonder whether the GAN-style discriminator is **stable** when expert data is extremely scarce. Have the authors looked into more robust density ratio estimation methods?

2. The ablations show that the best β depends on data quality, but users usually don't know this in advance. **Is there a way to set β adaptively?** This feels like an important practical gap.

3. The paper claims to remove the coverage assumption, but I'm curious whether the theoretical guarantees still hold **when the expert and suboptimal datasets cover completely disjoint state spaces**. This extreme case is not discussed.

**Limitations:**

The paper only briefly mentions limitations at the end of the conclusion. More discussion on discriminator instability, the fixed β under imbalanced dataset sizes, and extension to discrete action spaces would be appreciated.

**Strengths And Weaknesses:**

Strengths:
1. The mixture distribution decomposition provides a reasonable approach to addressing the coverage assumption issue, and the authors attempt to formally prove its equivalence to the original KL divergence minimization objective without introducing additional regularization terms. The two core propositions have complete derivations, which I find convincing.

2. The strict convexity of the Gumbel regression loss theoretically guarantees a unique global optimum. Compared to SMODICE and DemoDICE, the proposed objective also removes the need to sample from the initial state distribution, which is a nice practical benefit over SMODICE and DemoDICE.

3. The experiments cover 24 offline IL tasks, a limited coverage setting, and a maritime navigation task. The improvement over ReCOIL is notable, and the ablations are helpful for understanding each component's role.

Weaknesses:

1. The **readability of the paper** needs improvement. The abstract spends too much space on the limitations of prior work, leaving little room to describe the proposed method, which results in an unbalanced and overly long abstract. I would also suggest **including a method figure**, as the current text-only presentation makes it harder to follow the overall pipeline.

2. The second part of the related work section, "DICE-based Methods," only summarizes existing DICE methods **without clearly explaining how ReDICE differs from them**. It's a bit unclear to me what the key distinction is, especially compared to ReCOIL which also involves a mixture distribution.

3. ReDICE relies on a GAN-style discriminator for density ratio estimation, but I'm not sure how stable this is in practice. When there's only one expert trajectory, **the discriminator could easily overfit**, and this risk is not discussed at all in the paper.

4. The mixture coefficient β is fixed at 0.5 based on ablation results, but I find this **unconvincing without theoretical justification**. In practice the expert and suboptimal datasets can differ a lot in size, so assuming equal contributions seems a bit arbitrary.

---

> ### Author Rebuttal · Authors · 2026-03-31
>
> We thank reviewer for helpful comments. We provide our responses below to address your concerns.
>
> >**W1.** The readability of the paper needs improvement...
>
> We will revise the abstract to better balance prior limitations and our method description, and add a pipeline figure to improve clarity.
>
> >**W2.** The "DICE-based Methods" section mainly summarizes prior work and does not clearly distinguish ReDICE, particularly from ReCOIL.
>
> The "DICE-based Methods" section summarizes the general idea of DICE and its applications. We will revise it to explicitly introduce our contribution and highlight distinctions. Differences with ReCOIL are discussed in our response to **Weakness 1 of Reviewer iXLn**.
>
> >W3. When there's only one expert trajectory, GAN-based discriminator could easily overfit
>
> We follow prior DICE-based methods and use gradient penalty to stabilize training. In all experiments, expert dataset has only one expert trajectory and we do not observe instability, suggesting the discriminator remains stable in low-data regimes.
>
> > Q1. ...high variance on door and hammer tasks may be due to GAN-based discriminator...
>
> **We attribute the high variance mainly to the intrinsic difficulty of these manipulation tasks rather than discriminator instability.** Discriminator-free methods (e.g., ReCOIL) also exhibit high variance, indicating this is not specific to GAN-based estimation.
>
> We also evaluate Least-Squares Importance Fitting (LSIF), which is shown to outperform GAN-based method on density ratio benchmarks [1]:
>
> |Dataset|Env|LSIF Discriminator|GAN-based Discriminator|
> |-|-|-|-|
> |cloned+expert|pen|39.0±9.1|**88.8±8.8**|
> ||door|46.6±20.3|**92.2±2.5**|
> ||hammer|52.2±40.2|**104.9±10.1**|
> |human+expert|pen|**82.7±9.5**|**82.6±3.6**|
> ||door|87.1±8.6|**94.3±1.8**|
> ||hammer|35.6±26.5|**108.0±4.9**|
>
> While LSIF is competitive, it shows higher variance and worse performance in our setting, suggesting GAN-based objectives are more stable.
>
> >W4. ...$\beta$ is fixed at 0.5 based on ablation results... In practice, the expert and suboptimal datasets can differ a lot in size...
>
> >Q2. ...best $\beta$ depends on data quality... how to set $\beta$ adaptively?
>
> We clarify the following points:
>
> - **$\beta=0.5$ is not based on ablation** but follows common practice (e.g., ReCOIL or other IL works[2,3]).
>
> - **Our expert and suboptimal datasets already have a large size gap**. In our experimental setting, the expert dataset contains only one trajectory (up to 1K transitions), while the suboptimal dataset contains orders of magnitude more data (from 5K to over 1M see Table 3 in Appendix B.1).
>
> - **The best $\beta$ does not depend on data quality**. Our ablation (Section 5.4) shows $\beta=0.5$ performs best on average. The table below reports the best $\beta$ and gap (in %) to $\beta=0.5$.
>
>    At the task level, values such as 0.5 or 0.7 can be optimal in both high-quality (medium+few-expert) and low-quality (random+few-expert) settings, indicating no consistent pattern linking best $\beta$ to data quality. $\beta=0.5$ is optimal in most cases, with small gaps otherwise, supporting it as a robust default. **Thus, this supports our initial design choice of using $\beta=0.5$ as a fixed parameter without adaptive tuning.**
>
> |Dataset|Env|Best β|Gap(%)|
> |-|-|-|-|
> |random+expert|hopper|0.3|0.2|
> ||halfcheetah|**0.5**|0.0|
> ||walker2d|0.3|0.2|
> ||ant|**0.5**|0.0|
> |random+few-expert|hopper|**0.5**|0.0|
> ||halfcheetah|0.7|**4.2**|
> ||walker2d|0.7|0.1|
> ||ant|0.1|**9.0**|
> |medium+expert|hopper|**0.5**|0.0|
> ||halfcheetah|**0.5**|0.0|
> ||walker2d|**0.5**|0.0|
> ||ant|**0.5**|0.0|
> |medium+few-expert|hopper|**0.5**|0.0|
> ||halfcheetah|0.7|**4.5**|
> ||walker2d|**0.5**|0.0|
> ||ant|**0.5**|0.0|
>
> >**Q3.** ...whether guarantees hold when expert and suboptimal datasets have disjoint state spaces...
>
> We clarify two cases.
>
> - If supports are disjoint but share the same state-action space, our guarantee still holds. Since we have $d^{mix}(s,a)>0$ whenever $d^E(s,a)>0$, the KL decomposition remains well-defined even if $d^S$ assigns zero mass to expert states.
>
> - If datasets come from different MDPs, our theory does not apply, as ReDICE assumes shared dynamics. This becomes a cross-domain IL problem [4], requiring learned representations or domain mappings.
>
> **References:**
>
> [1]: Menon, Aditya, et al. "Linking losses for density ratio and class-probability estimation." ICML 2016.
>
> [2]: Al-Hafez, Firas, et al. "Ls-iq: Implicit reward regularization for inverse reinforcement learning." ICLR 2023.
>
> [3]: Sikchi, Harshit, et al. "A dual approach to imitation learning from observations with offline datasets." CoRL 2024.
>
> [4]: Fickinger, Arnaud, et al. "Cross-domain imitation learning via optimal transport." ICLR 2022.
>
> ---
>
> **We hope that our responses address your questions and concerns. We will incorporate all discussions and new experiments into the revised version. If there are any additional questions or comments, we would be happy to address them.**

---

> > ### Author Rebuttal · Reviewer_sgma · 2026-04-02
> >
> > We thank the authors for the detailed rebuttal. I respond to the remaining points below:
> >
> > > **W1: Readability.**
> >
> > **W1:** I appreciate the authors' willingness to revise the abstract and add a pipeline figure. Since I have **not yet seen the revised writing or the figure**, I am unable to evaluate whether these changes sufficiently address my readability concerns at this time.
> >
> > > **W3 & Q1: GAN Discriminator Stability.**
> >
> > **W3 & Q1:** This remains my **primary concern**. "Not observing instability" is **not equivalent to demonstrating stability**. The LSIF comparison shows GAN-based estimation has lower variance relatively, but this is a **comparison between methods**, not direct evidence of the discriminator's own training stability. Without such evidence (e.g., **training loss curves or gradient norm statistics** under the one-trajectory setting), I find the stability claim **insufficiently supported**.
> >
> > > **W4 & Q2: Fixed β = 0.5.**
> >
> > **W4 & Q2:** The ablation table is helpful and supports β = 0.5 as a **reasonable default**. However, the **9.0% gap** on `ant + random+few-expert` is **non-trivial**. The claim that "best β does not depend on data quality" is **overstated**. I hope the authors could further explain under what conditions this default might be suboptimal and how practitioners should handle such cases.
> >
> > The rebuttal addresses some of my concerns, but the issues above remain unresolved. I maintain my original score.

---

> > > ### Author Response · Authors · 2026-04-03
> > >
> > > Thank you for your valuable comments. We believe that we can address your remaining concerns with the following responses:
> > >
> > > >W1: Readability.
> > >
> > > We provide the pipeline figure summarizing our method at:
> > > https://anonymous.4open.science/r/icml2026-rebuttal-CD77/ReDICE_pipeline.pdf
> > >
> > > We were unable to include the revised abstract in the previous phase due to the 5000-character limit. Our revised abstract is as follows:
> > >
> > > Imitation Learning (IL) learns high-quality policies from expert demonstrations but degrades in low-expert-data regimes. To address this, recent work studies *offline IL with supplementary data*, augmenting expert data with trajectories from suboptimal policies. A prominent framework is Distribution Correction Estimation (DICE), which estimates density ratios via the dual of a divergence minimization problem between learned and expert visitation distributions. However, existing DICE-based methods either rely on a strict coverage assumption or introduce additional dataset regularization, limiting performance. We propose **ReDICE**, a new DICE-based method that addresses these issues through an objective-level reformulation. Our approach constructs a mixture-distribution objective that preserves the original expert-imitation objective while removing the coverage assumption, and its dual reduces to a stable Gumbel regression objective for efficient optimization. We further introduce a novel policy extraction mechanism that improves performance. Experiments on standard and real-world offline IL benchmarks show that ReDICE consistently outperforms prior methods and achieves state-of-the-art results.
> > >
> > > > W3 & Q1: GAN Discriminator Stability.
> > >
> > > We agree that comparing final performance does not directly demonstrate discriminator training stability, and we appreciate the suggestion to provide explicit training diagnostics.
> > >
> > > In our experiments, we actually monitor the discriminator training loss to assess whether the discriminator is stable. The training loss curves across 5 seeds on the hammer and door tasks are visualized in the plots provided at: https://anonymous.4open.science/r/icml2026-rebuttal-CD77/disc_training_plot.pdf
> > >
> > > Across all evaluated tasks, the loss decreases smoothly and consistently, without oscillation, divergence, or sudden spikes. The variance remains bounded throughout training, and the curves converge to stable plateaus after the initial phase. These observations indicate that the discriminator is well-trained and does not exhibit instability in the one-trajectory setting.
> > >
> > > > W4 & Q2: ...supports β = 0.5 as a reasonable default... I hope the authors could further explain under what conditions this default might be suboptimal and how practitioners should handle such cases.
> > >
> > > We agree that the 9.0% gap on ant + random+few-expert is non-trivial and represents an outlier case. We will revise our wording to clarify that our claim refers to the **lack of a consistent trend**, rather than strict independence from data quality.
> > >
> > > As shown in our previous response, $\beta=0.5$ is optimal in the majority of tasks, and in most remaining cases the performance gap is negligible (typically <0.3%). Only 3 tasks (ant+random+few-expert, halfcheetah+random+few-expert, halfcheetah+medium+few-expert) exhibit large gaps (4–9%), where $\beta=0.5$ is suboptimal. We cannot identify a common pattern among these cases, as they differ in both environment (halfcheetah vs. ant) and data quality (random+few-expert vs. medium+few-expert).
> > >
> > > In practice, it is difficult to identify optimal values for key hyperparameters (such as $\beta$) in offline IL/RL, which is a common issue observed across many prior works in this area. As shown in the table in our response to **W2 of Reviewer QSW3**, in the 3 outlier cases where $\beta=0.5$ is suboptimal, ReDICE with $\beta=0.5$ still improves over the best baselines by **21%–376%**. This large improvement indicates that, while $\beta=0.5$ is not always optimal, it still provides strong performance compared to prior works.
> > >
> > > Finally, while our design treats $\beta$ as a fixed parameter to avoid additional tuning complexity, we agree that practical guidance is important when $\beta=0.5$ does not yield the desired performance. Based on Figure 4 of our paper, $\beta$ in the range {0.3, 0.5, 0.7} provides the best overall performance. This is also consistent with the results in the table from our previous response, where the best $\beta$ across tasks always lies in this range, except for ant+random+few-expert, where $\beta=0.1$ performs best. Even in that case, $\beta=0.3$ performs nearly as well as $\beta=0.1$ (118.19 vs 118.82 according to Table 7 in our paper). **In practice, we recommend $\beta=0.5$ as a strong default, with optional tuning over {0.3, 0.5, 0.7} when higher performance is required.**
> > >
> > > ---
> > > **We hope our responses have satisfactorily addressed your remaining concerns. We will definitely incorporate all discussions and your suggestions into the revised version.**

---

### Official Review · Reviewer_ZM9E · 2026-03-12

**Soundness:** 2
**Presentation:** 4
**Significance:** 2
**Originality:** 2
**Overall Recommendation:** 4
**Confidence:** 4

**Summary:**

The paper presents ReDICE, a DICE-based method for offline-IL that aims to integrate an unlabeled dataset while avoiding the coverage assumption of previous methods. The main derivation closely follows SMODICE, but integrates the mixture distribution $d^\textit{mix}(s,a) = \beta d^{E} + (1-\beta) d^{S}$ into objective instead of unlabeled demonstrations $d^{S}$, that is, it exploits the identity $D\_\text{KL}(d^\pi(s,a)|| d^{E}(s,a)) = D\_\text{KL}(d^\pi(s,a)|| d^\textit{mix}(s,a)) + \mathbb{E}\_{d^\pi(s,a)} \log \frac{d^\textit{mix}(s,a))}{d^{E}(s,a)}$ to obtain a KL-regularized reward maximization problem. The resulting algorithm can be obtained by following techniques of related methods, namely the dual-V formulation to optimize the value function, and AWR with ODICE for policy extraction. For AWR the paper proposes to downscale $V(s)$ by a parameter $\tau \in [0,1]$. ReDICE outperforms strong baselines on standard offline-IL D4RL benchmark environments, and on a marine navigation task based on the NaviSIM simulator and real vessel trajectories.

**Compliance With Llm Reviewing Policy:**

Affirmed.

**Final Justification:**

The rebuttal provided useful additional experiments to evaluate the effect of $\tau$-scaling, which also demonstrated that this is not the only reason for the improved empirical results. This prompted me to increase the overall rating to 4.

Overall, the paper is still borderline due to its quite incremental nature.

**Key Questions For Authors:**

1. Can you provide the source code and NaviSIM training environment?
2. Could you explicitly clarify which parts of the derivations do not follow straightforwardly from prior work? Specifically, what unique challenges did the mixture-distribution objective introduce to the duality that were not present in previous formulations?
3. How do the baseline methods perform when tuning the temperature parameter $\tau$?

**Limitations:**

yes

**Strengths And Weaknesses:**

## Soundness
Using a mixture only for expanding the support of the unlabeled dataset is reasonable, as it does not change the optimal solution (in contrast to ReCOIL).
The theoretical results (which mostly follow prior work) are correct. Scaling the magnitude of the value function in AWR is theoretically justified.
The experimental results are consistent with the claims. The experimental design is reasonable and follows common praxis of related work.

However, the lack of supplementary code is a big caveat, as it harms reproducibility. It would also be useful to release the dataset and enviroment implementation of the NaviSIM experiment, so that future research can compare to ReDICE on this benchmark. Furthermore, since the introduction of $\tau$ made a significant effect on the performance of ReDICE, while not being tied to its specific derivations, it would important to also evaluate its effect on the related methods, e.g. ReCOIL.

## Presentation
The presentation is very good.
- The paper is well-written and overall polished. I
- I think that the preliminary section is well-suited to introduce readers that are less familiar with the close related work and the underlying theory.
- The algorithmbox is very useful for summarizing the resulting algorithm. If space was needed, the policy extraction could refer to Eq. 23.
- The supplementary material is reasonable and provides the most important implementation details and hyperparameters.

Minor: A period is missing on line 401.

## Significance
The empirical results on D4RL and NaviSIM are strong, making it at least somewhat interesting for researchers in the field. However, the paper's significance is attenuated by its incremental nature. The most impactful contribution may actually be the $\tau$-scaling. However, without an ablation showing how this scaling affects baselines like ReCOIL, it is difficult to disentangle the gains of the core algorithm from this specific hyperparameter tuning.

## Originality
- The paper does not seem to provide any compelling new insights.
- The advances seem very incremental compared to SMODICE and ReCOIL (which already used a mixture distribution to eliminate the coverage assumption, but also applied it to the policy distribution).
- The proofs and derivations in Appendix A seem to be mostly reworkings of the related work, applying the same derivation for the specific problem formulation. While mathematically sound, this makes it difficult to clearly distinguish its contribution from related literature.

---

> ### Author Rebuttal · Authors · 2026-03-30
>
> We thank the reviewer for the constructive comments and valuable feedback. Below, we provide our detailed responses to address your concerns.
>
> >**Q1.** Can you provide the source code and NaviSIM training environment?
>
> We will release the full implementation of ReDICE, including training scripts and hyperparameter configurations, upon acceptance. The NaviSIM environment [1] is publicly available at https://github.com/quanganh1999/ShipNaviSim, and our work builds directly on it. We follow the data pipeline of NaviSIM repo to process AIS data obtained from [MarineTraffic](https://www.marinetraffic.com/).
>
> > **Q2**. Could you explicitly clarify which parts of the derivations do not follow straightforwardly from prior work? ...
>
> We agree that applying Lagrangian duality to a divergence-regularized occupancy optimization follows the standard DICE template. However, our contribution is not the dualization procedure itself, but showing that a carefully designed mixture-based primal leads to a materially different and useful objective.
>
> Specifically, the nontrivial contributions are:
>
> * **Objective-level reformulation.** Our new DICE objective (Eq. (17)) preserves the original IL objective while removing the coverage assumption. This differs from prior work such as ReCOIL, which minimizes a divergence between mixture distributions. As a result, ReDICE maintains equivalence to the original expert-imitation objective, leading to both theoretical and practical advantages over ReCOIL (see our response to **Weakness 1 of Reviewer iXLn**).
>
> * **New dual characterization.** Propositions 4.1 and 4.2 show that this formulation induces a Gumbel regression objective together with a principled $\tau$-scaled policy extraction. To our knowledge, this connection does not appear in prior DICE-based methods and arises from our formulation.
>
> These components are essential to ReDICE and are validated in the ablation study (Section 5.4).
>
> We also appreciate the reviewer’s comment that $\tau$-scaling is an impactful component that can be applied to enhance other methods, such as ReCOIL. However, $\tau$-scaling alone does not fully explain the gains of ReDICE. As shown in Figure 3 (Section 5.4), while it consistently improves performance, its impact is smaller than that of the mixture formulation and Gumbel regression loss. This highlights that the overall performance gains arise from the combination of components, with the objective design playing a central role.
>
> Overall, our contribution is not an incremental modification, but an objective-level reformulation that **(i) removes the coverage assumption while preserving the IL objective, (ii) induces a new dual structure with improved optimization properties, and (iii) provides a principled policy extraction mechanism.** We will revise the paper to clarify these distinctions.
>
> >**Q3.** How do the baseline methods perform when tuning the temperature parameter $\tau$?
>
> We evaluate ReCOIL with $\tau$-scaling by replacing its policy extraction and tuning $\tau \in$ {0.5,0.7,0.9,1.0}. We observe consistent trends with ReDICE: For ReCOIL also, $\tau=0.9$ performs best on locomotion and cloned tasks, while $\tau=1.0$ works better on kitchen and human datasets.
>
> Results below (mean over 5 seeds) show that $\tau$-scaling consistently improves ReCOIL (≈20% relative gain). However, ReCOIL w/ $\tau$-scaling still underperforms ReDICE on most tasks, with ≈20% average gap, indicating that improvements mainly come from the core objective rather than $\tau$ alone.
>
> |Dataset|Env|ReCOIL w/ τ-scaling|ReCOIL|ReDICE|
> |---|---|---|---|---|
> |random+expert|hopper|102.79|98.33|**109.80**|
> ||halfcheetah|90.94|78.78|**91.82**|
> ||walker2d|71.79|105.72|**107.98**|
> ||ant|126.07|116.43|**126.32**|
> |random+few-expert|hopper|90.15|86.65|**93.63**|
> ||halfcheetah|37.44|5.25|**79.31**|
> ||walker2d|34.05|12.59|**108.26**|
> ||ant|96.71|56.68|**108.15**|
> |medium+expert|hopper|110.84|85.25|**111.01**|
> ||halfcheetah|75.91|78.88|**91.80**|
> ||walker2d|108.40|108.27|**108.40**|
> ||ant|123.02|119.21|**123.24**|
> |medium+few-expert|hopper|92.59|38.88|**109.50**|
> ||halfcheetah|57.85|41.90|**81.57**|
> ||walker2d|107.86|71.44|**108.70**|
> ||ant|108.79|104.03|**111.73**|
> |cloned+expert|pen|40.80|45.59|**88.84**|
> ||door|87.61|16.18|**92.23**|
> ||hammer|31.65|2.60|**104.86**|
> |human+expert|pen|84.35|**85.04**|82.62|
> ||door|92.36|84.02|**94.31**|
> ||hammer|80.41|86.45|**107.97**|
> |partial+expert|kitchen|**54.25**|48.80|54.00|
> |mixed+expert|kitchen|49.17|**49.35**|48.25|
> |Avg||81.49|67.76|**97.68**|
>
> **References:**
>
> [1]: Pham, Quang Anh, Janaka Chathuranga Brahmanage, and Akshat Kumar. "ShipNaviSim: Data-Driven Simulation for Real-World Maritime Navigation." AAMAS 2025.
>
> ---
>
> **We hope that our responses address your questions and concerns. We will incorporate all discussions and new experiments into the revised version. If there are any additional questions or comments, we would be happy to address them.**

---

> > ### Author Rebuttal · Reviewer_ZM9E · 2026-04-01
> >
> > The authors added an important ablation to test the effect of $\tau$-scaling on related work.

---

### Official Review · Reviewer_QSW3 · 2026-03-13

**Soundness:** 3
**Presentation:** 3
**Significance:** 3
**Originality:** 3
**Overall Recommendation:** 4
**Confidence:** 3

**Summary:**

This paper proposes ReDICE, a method that addresses the limitations of offline imitation learning (IL) with suboptimal supplementary data. It identifies that existing Distribution Correction Estimation (DICE) methods either rely on a restrictive assumption that suboptimal data covers all expert visitations or introduce regularizations that distort the main IL objective.  ReDICE minimizes the KL divergence between the learned policy's occupancy measure and a mixture of expert and suboptimal data distributions, completely bypassing the coverage assumption. The authors show that the dual of this new objective is equivalent to a strictly convex Gumbel regression loss, thereby improving optimization stability. They also introduce a policy extraction mechanism with a tunable temperature parameter to mitigate value overestimation.  Experiments on D4RL benchmarks and a maritime navigation simulator demonstrate that ReDICE outperforms current baselines, especially when expert coverage is low.

**Compliance With Llm Reviewing Policy:**

Affirmed.

**Final Justification:**

This paper proposes ReDICE, a method that addresses key limitations of offline imitation learning (IL) with suboptimal supplementary data. The work identifies shortcomings in existing DICE-based methods and introduces a formulation that minimizes the KL divergence between the learned policy’s occupancy measure and a mixture of expert and suboptimal data distributions, thereby bypassing the restrictive coverage assumption. The proposed idea is interesting, and the empirical results are solid and generally convincing.

The authors’ rebuttal has addressed my main concerns and provided helpful clarifications. In particular, the discussion on statistical significance, discriminator robustness, and hyperparameter sensitivity helps better contextualize the empirical results and the practical applicability of the method.

While some limitations remain, they do not materially affect my overall assessment of the quality and novelty of this work. I therefore maintain my recommendation of Weak Accept (4).

**Key Questions For Authors:**

1. 1.	Can the "orthogonal gradient update" be applied to other DICE-based methods, or is its effectiveness unique to the Gumbel loss formulation?
2. Since Eq. 21 and Eq. 24 are only strictly equivalent under deterministic transitions, how biased might the empirical Gumbel objective become in environments with significant transition stochasticity? Have the authors evaluated ReDICE in such settings?
3. The ablation study shows that performance is sensitive to the temperature parameter τ. In a strictly offline setting where interaction with the environment is not allowed, is there a validation metric, heuristic, or proxy objective that could help practitioners tune τ?
4. In the maritime navigation experiment, actions are reconstructed from AIS state trajectories using an inverse kinematics model. How sensitive is this reconstruction to noise in the real-world logs, and could such noise affect the discriminator’s ability to distinguish expert trajectories from the constructed suboptimal data?

**Limitations:**

Yes

**Strengths And Weaknesses:**

Strengths
1. The paper proposes an interesting KL divergence formulation for offline IL based on a mixture distribution that naturally circumvents the restrictive expert coverage assumption without relying on auxiliary regularization terms.

2. The paper also introduces a practical policy extraction mechanism utilizing a temperature scalar to mitigate the empirical issue of weight collapse caused by value overestimation.

3. The empirical validation demonstrates improved performance across diverse D4RL continuous control tasks and successfully applies the algorithm to a maritime navigation dataset (ShipNaviSim).

Weaknesses
1. Hyperparameter Sensitivity: The policy extraction mechanism relies on a temperature parameter τ. Ablation results show that performance is sensitive to this parameter and requires per-task tuning. The method does not provide a principled way to adaptively determine this parameter in a purely offline setting.

2. Dependence on Discriminator Quality: The reward function r(s, a) is derived from a learned discriminator trained to distinguish expert and suboptimal data. Errors or overfitting in the discriminator may introduce bias into the reward estimation, which could propagate into the Gumbel regression objective and the subsequent policy extraction step.

3. Statistical Significance of Gains: Although the overall performance improvements are strong, on several individual tasks, the performance overlaps with strong baselines such as ReCOIL or DemoDICE within one standard deviation. It would be helpful to clarify whether the observed gains are statistically significant at the task level.

---

> ### Author Rebuttal · Authors · 2026-03-30
>
> We thank the reviewer for the helpful comments. We address the concerns below.
>
> >**W1.** Discriminator overfitting may bias reward estimation.
>
> While discriminator overfitting is possible, there are many ways to address this issue. For example, regularization methods like Dropout or Gradient Penalty are effective in reducing overfitting on MuJoCo locomotion and manipulation tasks [1]. For more complex settings, stronger generative models (e.g., diffusion [2], flow matching [3]) can further improve robustness.
>
> >**W2.** Gains may not be statistically significant.
>
> We quantify improvements using percentage gap (Gap) relative to the strongest baseline. ReDICE achieves >5% gains in most tasks, especially in low-quality regimes. Negative gaps are rare (4 cases total), with only one moderate drop (-6.2%); others are minor. This indicates consistent, non-random improvements.
> |Dataset|Env|Gap(%)|
> |-|-|-|
> |random+expert|hopper|**5.5**|
> ||halfcheetah|**12.0**|
> ||walker2d|0.1|
> ||ant|-0.4|
> |random+few-expert|hopper|-6.2|
> ||halfcheetah|**376.9**|
> ||walker2d|1.3|
> ||ant|**21.4**|
> |medium+expert|hopper|**30.2**|
> ||halfcheetah|**21.4**|
> ||walker2d|0.1|
> ||ant|3.4|
> |medium+few-expert|hopper|**77.6**|
> ||halfcheetah|**72.1**|
> ||walker2d|**31.5**|
> ||ant|**7.4**|
> |cloned+expert|pen|**94.9**|
> ||door|**469.9**|
> ||hammer|**2344.1**|
> |human+expert|pen|-2.9|
> ||door|**12.2**|
> ||hammer|**24.9**|
> |partial+expert|kitchen|**10.7**|
> |mixed+expert|kitchen|-2.2|
>
> >**Q1.** Is orthogonal gradient update generally applicable?
>
> Yes, it can be applied to any DICE method. Specifically, we compute the forward and backward gradients by applying the stop-gradient operator on $V(s')$ and $V(s)$, respectively. The orthogonal gradient (ort_grad) is then obtained by combining the forward gradient with the projected backward gradient.
>
> However, its effectiveness depends on the objective. Applying orthogonal gradient alone (+ort_grad) generally degrades SMODICE performance. Moreover, combining it with Gumbel regression loss and $\tau$-scaled policy extraction (+all) further worsens the results:
> |Method|SMODICE|+ort_grad|+all|
> |-|-|-|-|
> |Avg return|**53.25**|33.39|5.25|
>
> Thus, orthogonal gradient updates must align with our objective design.
>
> >**Q2.** Effect of stochastic transitions?
>
> Handling stochastic transitions in offline IL is difficult due to unknown dynamics. Learning a transition model $\hat T(s'|s,a)$ is possible but introduces approximation error depending on data coverage. This requires designing a robust IL approach [5] to deal with the mismatch between $\hat T$ and the actual environment transition function.
>
> Because prior offline IL works focus on deterministic MuJoCo tasks, conducting additional experiments on stochastic environments is challenging, as it requires recalibrating the algorithm and rerunning baselines within limited rebuttal time. We leave this extension for future work.
>
> >**Q3.** How to tune $\tau$ offline?
>
> Tuning policy temperature is standard in offline methods (e.g., IQL [6], XQL [7]). Our strongest baseline, ReCOIL, further requires tuning two hyperparameters: the policy temperature $\alpha$ and $\lambda$, which controls the regularization strength in the extreme value learning (See Appendix F.1 of ReCOIL).
>
> In contrast, ReDICE requires smaller hyperparameter tuning. As stated in Appendix B.3, we fix $\tau = 0.9$ for most tasks, while $\tau = 1.0$ performs better only in few cases (Kitchen and human+expert tasks). Therefore, we adopt the commonly accepted assumption of having limited evaluation access to the environment.  $\tau$ then can be selected from a very small candidate set {0.9, 1.0} requiring at most two evaluation trials.
>
> >**Q4.** Sensitivity to AIS noise?
>
> Noise is mitigated because (i) AIS trajectories are smooth and preprocessed (interpolation/filtering), and (ii) both expert and suboptimal data share the same reconstruction pipeline, avoiding bias. Empirically, we observe stable training and consistent improvements. Incorporating noise-aware models is a promising future direction.
>
> **References:**
>
> [1]: Orsini, Manu, et al. "What matters for adversarial imitation learning?." NeurIPS 2021.
> [2]: Lai, Chun-Mao, et al. "Diffusion-reward adversarial imitation learning." NeurIPS 2024.
> [3]: Wan, Zhenglin, et al. "FM-IRL: Flow-Matching for Reward Modeling and Policy Regularization in Reinforcement Learning." arXiv preprint arXiv:2510.09222.
> [4]: Panaganti, Kishan, et al. "Distributionally robust behavioral cloning for robust imitation learning." CDC 2023.
> [5]: Kostrikov, Ilya, Ashvin Nair, and Sergey Levine. "Offline reinforcement learning with implicit q-learning." ICLR 2022.
> [6]: Garg, Divyansh, et al. "Extreme q-learning: Maxent rl without entropy." ICLR 2023.
>
> ---
>
> **We hope that our responses address your questions and concerns. We will incorporate all discussions and new experiments into the revised version. If there are any additional questions or comments, we would be happy to address them.**

---

> > ### Author Rebuttal · Reviewer_QSW3 · 2026-04-04
> >
> > Thanks for the detailed rebuttal. I am maintaining my positive rating of this paper.

---

### Official Review · Reviewer_iXLn · 2026-03-13

**Soundness:** 2
**Presentation:** 3
**Significance:** 2
**Originality:** 2
**Overall Recommendation:** 4
**Confidence:** 3

**Summary:**

This paper proposes ReDICE, a novel approach for offline imitation learning (IL) with suboptimal datasets. The core contribution lies in relaxing the strict coverage assumption required by existing DICE methods through a mixture distribution technique, and proving that the dual of this objective is equivalent to a Gumbel regression loss, thereby guaranteeing optimization stability. Additionally, the authors introduce a temperature-parameterized policy extraction mechanism to mitigate value overestimation. Experiments on D4RL tasks and a real-world maritime navigation scenario validate the method, achieving SOTA performance under limited expert coverage settings.

**Compliance With Llm Reviewing Policy:**

Affirmed.

**Final Justification:**

Most of my concerns have been addressed; however, in light of the shared concerns among the reviewers regarding the manuscript’s limitations in significance and originality, I have decided to maintain my original score.

**Key Questions For Authors:**

see Weaknesses

**Limitations:**

yes

**Strengths And Weaknesses:**

Strengths

- The manuscript is well-organized and clearly written.

- Providing a theoretical solution to the numerical instability issues that have long plagued DICE-based methods.

- The method is validated on a maritime navigation task using AIS data. This goes beyond standard MuJoCo benchmarks and showcases strong practical applicability.

- The experimental design covers a diverse set of scenarios.

Weaknesses

1. The theoretical distinction from concurrent work ReCOIL (which also addresses coverage issues) is not sufficiently pronounced, lacking in-depth comparison regarding convergence speed or sample efficiency.

2. The method involves a three-stage pipeline (discriminator training, value function optimization, and policy extraction), incurring significantly higher computational overhead than end-to-end methods (e.g., BC or DWBC). However, the paper provides no analysis of training time or computational resource comparison, limiting the assessment of its practicality in resource-constrained scenarios.

3. The use of an empirical Bellman operator (single-sample s′) to approximate the true transition dynamics (Eq. 24) introduces theoretical approximation errors. Although following prior work, the paper lacks sufficient discussion on the trade-off between strict theoretical guarantees and implementation convenience.

---

> ### Author Rebuttal · Authors · 2026-03-30
>
> We thank reviewer for your helpful comments and feedback. We provide our responses below to address your concerns.
>
> >**W1**. Limited distinction from ReCOIL, with insufficient comparison on convergence or sample efficiency.
>
> We agree that ReCOIL also addresses coverage issues, but its objective differs from ours in an important way. Let $d^{mix}(s,a)=\beta d^E(s,a)+(1-\beta)d^S(s,a)$. Consider the three objectives:
>
> Original IL objective:
> $$ \min_{d^{\pi}} D_f\big(d^{\pi}(s,a)||d^E(s,a)\big) \qquad(1)$$
> ReCOIL objective:
> $$ \min_{\pi,d} D_f\big(\beta d(s,a)+(1-\beta)d^S(s,a) || d^{mix}(s,a)\big) \qquad(2)$$
> $$\text{s.t.}\quad d(s,a)=(1-\gamma)d_0(s)\pi(a|s)+\gamma \sum_{s',a'}d(s',a')\mathcal{P}(s|s',a')\pi(a|s)$$
> ReDICE objective:
> $$\min_{d} E_{d(s,a)}\big[\log\frac{d^{mix}(s,a)}{d^E(s,a)}\big] + D_{KL}(d(s,a)||d^{mix}(s,a)\big) \qquad (3)$$
> $$\text{s.t.}\quad\sum_{a} d(s,a)=(1-\gamma)d_0(s)+\gamma \sum_{s',a'}d(s',a')\mathcal{P}(s|s',a')$$
>
> The constraints in both (2) and (3) are Bellman flow constraints to ensure that $d(s,a)$ is a valid occupancy measure $d^{\pi}(s,a)$ in (1). From Eqs. (1)–(3), ReDICE has three key advantages over ReCOIL:
>
> - **Preserves the original objective while relaxing coverage.** ReCOIL matches divergences between mixture distributions that only share the same optimum as (1), whereas ReDICE remains exactly equivalent to (1) under KL.
>
> - **Avoids adversarial max–min optimization.** The dual formulation of ReCOIL solves a max–min problem over $Q$ and $\pi$ (Theorem 1 in [1]), which may induce extrapolation errors from out-of-distribution actions in offline settings [2]. ReDICE reduces to a minimization over $V$, avoiding this issue.
>
> - **Simpler and more theory-aligned optimization.** To mitigate the OOD issues discussed above, ReCOIL relies on an in-sample surrogate objective to jointly learn $Q(s,a)$ and $V(s)$ (Eqs. (11)–(12) in [1]). This introduces an additional hyperparameter $\tau$ that must be tuned per task (Appendix F.1 of [1]). As a result, the practical implementation of ReCOIL creates a gap between theory and practice and increases tuning complexity. In contrast, ReDICE uses a single minimization over $V$, reducing tuning complexity.
>
> >**W2.** Higher computational cost than end-to-end methods like DWBC due to a three-stage pipeline, without analysis of training time or resources.
>
> The main cost of our approach is joint training of value and policy networks. The discriminator is lightweight (5,000 iterations, ~5 minutes). We report the total training time on a RTX 3090 GPU and computational requirements of different methods below:
>
> | Method|Time (minutes)|Neural network components |
> |-|-|-|
> |DWBC|46| $c$, $\pi$ |
> |SMODICE|106|$c$,$\pi$,$V$|
> |ReDICE| 151 | $c$, $\pi$, 2×$V$ |
> |ReCOIL| –   | $\pi$, $V$, 2×$Q$ |
>
> DWBC is the most efficient since it avoids value learning. ReDICE incurs additional cost compared to SMODICE due to double value learning, which improves stability [3].
>
> ReCOIL has a similar computational structure, but it is implemented in JAX, making direct runtime comparison with our PyTorch implementation less meaningful. Due to this difference in frameworks, we do not report training time for ReCOIL. Nevertheless, its computational requirements are comparable to ReDICE in terms of the number of learned components.
>
> >**W3.** Empirical Bellman operator introduces approximation error, with no discussion of theory–practice trade-offs.
>
> We agree that using the empirical Bellman operator in Eq. (24) introduces approximation error. This discrepancy arises primarily in stochastic environments, as Eq. (21) and Eq. (24) are strictly equivalent under deterministic transitions.
>
> Handling stochastic transitions in offline IL is inherently difficult due to unknown dynamics. Without environment interaction, unbiased estimates are unavailable. Learning a transition model $\hat T(s'|s,a)$ is possible but depends on data coverage and may introduce further errors.
>
> Thus, we follow prior DICE-based methods using empirical Bellman operators for tractability. This limitation is shared across offline IL methods. Moreover, standard benchmarks (e.g., D4RL) are deterministic, making this effect less prominent.
>
> Extending ReDICE to handle stochastic environments (e.g., via learned models or uncertainty-aware objectives) is an important future direction, which we will discuss in the revised version.
>
> **References:**
>
> [1]: Sikchi, Harshit, et al. "Dual rl: Unification and new methods for reinforcement and imitation learning." ICLR 2024.
>
> [2]: Kumar, Aviral, et al. "Conservative q-learning for offline reinforcement learning." NeurIPS 2020.
>
> [3]: Fujimoto, Scott, et al. "Addressing function approximation error in actor-critic methods." ICML 2018.
>
> ---
>
> **We hope that our responses address your questions and concerns. We will incorporate all discussions and new experiments into the revised version. If there are any additional questions or comments, we would be happy to address them.**

---

> > ### Author Rebuttal · Reviewer_iXLn · 2026-04-02
> >
> > I appreciate the authors' rebuttal, and my concerns have been largely addressed. However, considering the shared concerns of other reviewers regarding the inherent limitations in the manuscript's significance and originality , I have decided to maintain my original score.

---

### Decision · Program_Chairs · 2026-04-30

**Decision:**

Accept (regular)

**Comment:**

This work considers imitation learning in settings with few expert demonstrations. Unfortunately, many existing imitation learning methods degrade in such settings in both theory/practice and/or have strong assumptions. DICE-based policy estimation requires strong coverage of the demonstration distribution. This paper seeks to avoid these assumptions under DICE while optimizing expert imitation. It reformulates the KL divergence using a mixture distribution combining expert/suboptimal distributions to derive a DICE-based objective. Experimental results show significant improvements over baseline methods. An additional baseline shows that the improvement in performance over ReCOIL, which has similar motivations, is not simply due to a hyperparameter introduces in the proposed approach.

Overall, though someone incremental in the family of DICE methods, this is a solid contribution with good theory and experimental support. I recommend acceptance.